# Predicting the risk and speed of drug resistance emerging in soil-transmitted helminths during preventive chemotherapy

**Luc E. Coffeng** [1] ✉, **Wilma A. Stolk** [1] & **Sake J. de Vlas** [1]

Control of soil-transmitted helminths relies heavily on regular large-scale deworming of high-risk groups (e.g., children) with benzimidazole derivatives. Although drug resistance has not yet been documented in human soil-transmitted helminths, regular deworming of cattle and sheep has led to widespread benzimidazole resistance in veterinary helminths. Here we predict the population dynamics of human soil-transmitted helminth infections and drug resistance during 20 years of regular preventive chemotherapy, using an individual-based model. With the current preventive chemotherapy strategy of mainly targeting children in schools, drug resistance may evolve in soil-transmitted helminths within a decade. More intense preventive chemotherapy strategies increase the prospects of soil-transmitted helminths elimination, but also increase the speed at which drug efficacy declines, especially when implementing community-based preventive chemotherapy (population-wide deworming). If during the last decade, preventive chemotherapy against soil-transmitted helminths has led to resistance, we may not have detected it as drug efficacy has not been structurally monitored, or incorrectly so. These findings highlight the need to develop and implement strategies to monitor and mitigate the evolution of benzimidazole resistance.

Soil-transmitted helminths (STH) constitute a diverse group of nematode species that are transmitted via faecal contamination of the environment and infect approximately one billion people globally[1]. STH infections cause intestinal blood loss, iron deficiency anaemia, and protein malnutrition, especially in heavily infected individuals and high-risk populations with low iron reserves such as children and women of reproductive age[2]. The World Health Organization (WHO) has therefore targeted STH to be controlled in high-risk populations by 2030[3]. Long-term control or even elimination of STH is expected to require improved access to water, hygiene and sanitation[4,5]. In the short term, control of STH relies heavily on regular large-scale deworming of high-risk groups with mebendazole or albendazole, which are both benzimidazole derivatives[4]. Deworming is implemented as preventive chemotherapy (PC), meaning that all treatment-eligible individuals in endemic populations are offered treatment, without individual diagnosis. As such, the development of

benzimidazole resistance would present a severe threat to the effectiveness of control programmes and the health gains achieved thus far[6]. The risk of benzimidazole resistance among STH is considered to be real[7–10], primarily because benzimidazole resistance is widespread in intestinal helminths of small ruminants[11,12] and cattle[13], which are also controlled by means of regular deworming. For STH in humans, sub-optimal responses and genetic selection effects have been reported[7,14–17], but without conclusive evidence of resistance.

In veterinary nematodes, benzimidazole resistance has been linked to three single-nucleotide polymorphisms (SNPs) at codons 167, 198, and 200 of the beta-tubulin isotype 1 gene[11]. These canonical SNPs are widespread in veterinary intestinal helminths, even in populations that have not experienced selective pressure from drug treatment[18]. However, these known SNPs do not explain all variation in benzimidazole efficacy within and between nematode species[19], and the mode of inheritance varies between species (e.g., recessive vs. (co-)

[1]Department of Public Health, Erasmus MC, University Medical Center Rotterdam, Rotterdam, The Netherlands. ✉e-mail: l.coffeng@erasmusmc.nl

dominant)[11]. A study of ovine intestinal nematodes further suggested that multiple loci are involved in benzimidazole resistance as selection seemed to occur through a mix of soft and hard selective sweeps, which are selection events that result in parasite populations with normal or low genetic variation, respectively[20]. This suggests that mutations in one or a few loci may be essential for resistance to develop, but that additional mutations elsewhere in the genome also contribute towards resistance, though to a smaller degree per mutation. So far, based on studies in equines and ruminants, there is no clear evidence for loss of fitness in benzimidazole-resistant strains of intestinal nematodes[21,22].

Demonstrating the existence of benzimidazole resistance in human STH has proven challenging because estimates of drug efficacy are easily confounded by factors such as drug formulation and quality[7], and variation in study protocols and diagnostic techniques[7,23,24], but also biological factors such as inter-individual variation in benzimidazole pharmacokinetics[7], variation in pre-treatment infection levels[25-27], and density-dependent worm fecundity[28]. For instance, it remains uncertain whether reports of reduced efficacy of albendazole against hookworms[14] represent genuine cases of resistance; in a follow-up study, albendazole efficacy was found to be associated with the timing of treatment after the last meal and nutritional factors[29]. Detecting true drug resistance is further complicated by the fact that the current WHO-recommended survey design for evaluating drug efficacy[8], which is based on selection of egg-positive individuals before treatment, leads to overestimation of drug efficacy[30,31]. So far, the only data suggestive of selective pressure by benzimidazole derivatives is the observation of a SNP at codon 200 of the beta-tubulin gene in *Trichuris trichiura* in areas of Haiti and Kenya that have been subject to annual treatment with albendazole and diethylcarbamazine against lymphatic filariasis[16]. The SNP was present in 2008–2009 and significantly increased in frequency after a single round of treatment with albendazole. Similarly, the frequency of a SNP at codon 167 in *Ascaris lumbricoides* increased significantly in the Haitian site[16]. In both cases, the changes in genotype frequencies were suggestive of a co-dominant inheritance pattern (i.e., stable or increasing frequency of heterozygous genotypes after treatment). However, this could not be confirmed at the phenotypical level as drug efficacy was not evaluated. Still, benzimidazole resistance in STH may also be driven by yet unknown SNPs in loci other than the beta-tubulin gene, as in veterinary helminths[19,20] and as demonstrated for drug-induced benzimidazole resistance in laboratory-reared *Ancylostoma ceylanicum* in hamsters[32], a canine/feline hookworm species that can also cause patent infection in humans[33,34].

Mathematical models for drug resistance in veterinary helminths[35–43] have illustrated that development and spread of drug resistance is influenced by many factors related to worm biology (transmission cycle, parasite lifespan); treatment regimen (single *vs.* multiple drugs, dosage, frequency, efficacy); the proportion of the parasite population that is exposed to the treatment; and how resistance is genetically encoded, e.g., via a single locus (monogenic) or multiple loci (polygenic) in the worm genome. However, the case of human STH is somewhat different from the veterinary case, particularly in terms of treatment frequency (lower in humans) and the proportion of the population targeted by deworming programmes (also lower as children are primarily targeted). In this study, we investigated the risk and speed of drug resistance in human STH populations under selective pressure from PC. For this purpose, we use the first-ever individual-based model for evolution of monogenic and polygenic drug resistance in human helminth infections, which we developed as an extensively documented open-source software package (Supplementary Information A). We characterised the model for *Ascaris lumbricoides* and *Necator americanus* and explore the evolution of drug resistance in various PC scenarios and under various assumptions about the genetic mechanisms behind resistance.

## Results

For each of the two parasite species, *A. lumbricoides* and *N. americanus*, we simulated a database of 1000 random pre-PC population states with random transmission conditions, which represented a range of 10–80% baseline prevalence of female worm infection (up to 60% if measured by a single Kato-Katz faecal smear; Supplementary Information B, Supplementary Fig. B5). Next, for each of these baseline settings, we simulated the impact of 20 years of PC, assuming various combinations of PC target population and frequency, and the genetic mechanisms behind drug resistance in STH. In Fig. 1, we provide an example of model predictions for a scenario for community-based PC (treatment of all ages ≥2) against *N. americanus* with monogenic resistance and co-dominant inheritance. Compared to simulations in which we assumed absence of a resistance mechanism (top row of panels), co-dominant resistance (bottom row) led to a lower probability of elimination (37.9% vs. 65.3%), a rebound in infection levels, and a decline in drug efficacy to <50% within 5 to 10 years. Eventually, drug resistance dropped to 9.5%, which was the maximum level of monogenic resistance in the population where drug efficacy against each and every worm was reduced by 90%. Supplementary Information C holds similarly detailed results for all combinations of parasite species, PC strategy, and inheritance mechanism. From here on, we will mostly focus on the probability of elimination within 20 years and the average trends in drug efficacy in simulations that did not achieve elimination (red lines).

In general, the probability of elimination was higher when PC was implemented biannually (vs. annually) or community-based (vs. school-based, targeting only children of age 2–15; Table 1). Prospects of elimination decreased markedly depending on the assumed genetic mechanism underlying resistance (in descending order of probability of elimination): polygenic resistance (low or high phenotypic variation), monogenic recessive, monogenic co-dominant, monogenic dominant. In case of school-based PC, the probability of elimination was higher for *A. lumbricoides* than for *N. americanus*. This can be readily explained by the fact that *A. lumbricoides* populations primarily reside in the PC target group (children) whereas *N. americanus* infection levels continue to increase with age. For the same reason, community-based PC led to a larger jump in probability of elimination (compared to school-based PC) for *N. americanus* than for *A. lumbricoides*.

In case elimination was not achieved, the speed at which drug efficacy declined varied greatly between PC strategies and inheritance mechanisms (Fig. 2; lines correspond to the averages in Fig. 1). For both parasite species, drug efficacy declined more quickly with biannual PC and/or community-based PC. Especially for *Necator americanus*, the decline was faster under community-based PC than school-based PC (red vs. black lines). Drug efficacy also declined faster when resistance was inherited in a monogenic fashion, particularly if co-dominant or dominant. In case of recessive inheritance, the onset of the decline in drug efficacy was generally later and more variable across simulations (see Supplementary Information C for variation between individual simulations). The decline in drug efficacy was slowest in case of polygenic inheritance of resistance, particularly if drug efficacy was less variable (i.e., lower trait variation). In case of school-based PC, resistance evolved more quickly in *A. lumbricoides* than in *N. americanus*, due to higher selective pressure as for the former species, a larger proportion of worm population reside in children. Also, the average adult lifespan of *A. lumbricoides* is shorter (-1 year) than that of *N. americanus* (-3 years), making the genetic turn-over of *A. lumbricoides* populations higher such that resistant phenotypes more quickly replace susceptible phenotypes.

In addition to the PC strategy and inheritance mechanism of drug resistance, transmission conditions were an important determinant of achieving elimination within 20 years. The probability of elimination decreased with higher degrees of inter-individual variation in exposure

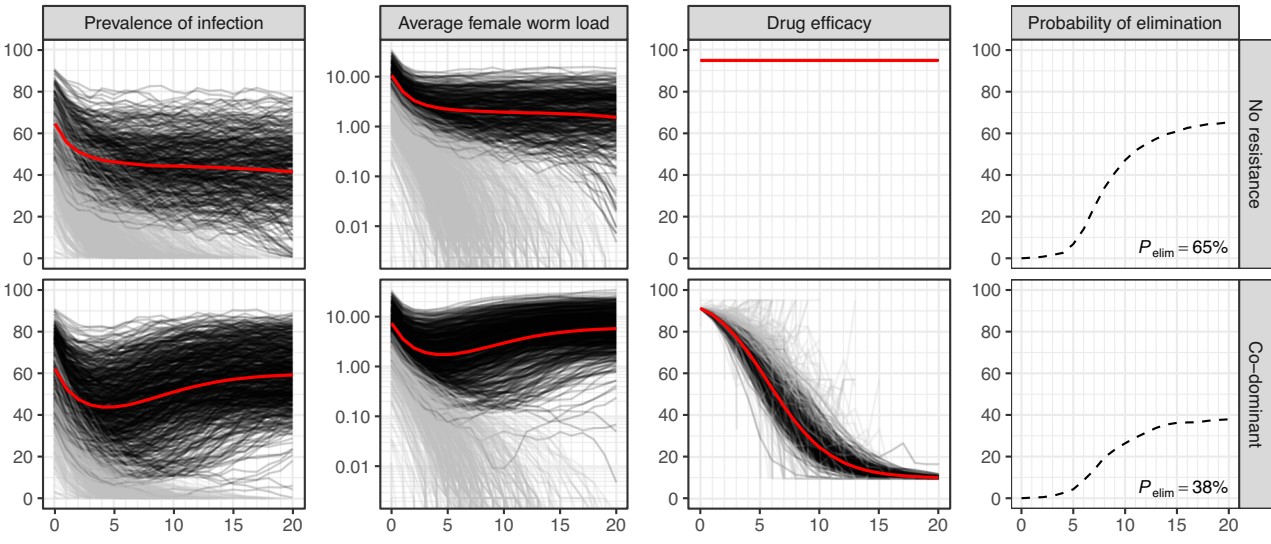

**Fig. 1 | Example of model-predicted population dynamics of *Necator americanus* during annual community-based preventive chemotherapy at 70% coverage, given illustrative scenarios with and without evolution of drug resistance.** Drug resistance is considered to be either absent (top row) or monogenic with co-dominant inheritance (bottom row). Thin lines represent single simulations (*N* = 1000), where each line is based on one simulated data point per year, timed right before a PC round; for an example of how infection levels fluctuate between PC rounds, see Supplementary Information A. A grey line colour indicates that elimination was achieved (i.e., zero female worms left within 20 years of PC) and a black line colour indicate ongoing transmission. Thick red lines are the average (arithmetic mean) of simulations in which elimination was not achieved. Infection levels are shown in terms of the percentage of people with at least one female worm (first column) and the average number of female worms per person (second column). Drug efficacy is defined as the probability of a worm being killed, averaged over the worm population (third column). The right-most column of panels summarises the proportion of simulations that have achieved elimination over time, as well as the final proportion after 20 twenty years (*P*elim, corresponding with results in Table 1). See Supplementary Information C for results of all combinations of PC strategy, inheritance mechanism, and parasite species.

**Table 1 | Model-predicted probability of elimination of *Ascaris lumbricoides* and *Necator americanus* in various scenarios for preventive chemotherapy (PC) and the genetic mechanisms behind drug resistance**

| PC strategy (coverage) | PC frequency | Probability of elimination (zero worms within 20 years of PC) | | | | | |
|---|---|---|---|---|---|---|---|
| | | | Polygenic (*h*² = 0.3)ᵃ | | Monogenic | | |
| | | No resistance mechanism | Low trait variation | High trait variation | Recessive | Co-dominant | Dominant |
| *Ascaris lumbricoides* | | | | | | | |
| School-based (95%) | Annual | 14 | 15 | 14 | 13 | 12 | 10 |
| | Biannual | 25 | 24 | 23 | 24 | 20 | 14 |
| Community-based (70%) | Annual | 35 | 34 | 34 | 33 | 26 | 16 |
| | Biannual | 64 | 64 | 63 | 62 | 42 | 23 |
| *Necator americanus* | | | | | | | |
| School-based (95%) | Annual | 5 | 5 | 5 | 6 | 5 | 5 |
| | Biannual | 6 | 6 | 6 | 7 | 7 | 6 |
| Community-based (70%) | Annual | 65 | 65 | 64 | 61 | 38 | 18 |
| | Biannual | 90 | 90 | 88 | 79 | 54 | 24 |

Probabilities are expressed as percentages (%) and are based on 1000 simulations that represent a mix of random transmission conditions. For each parasite species, the estimated probabilities for each PC and genetics scenario (i.e., each cell) are based on exactly the same random transmission conditions. School-based PC is assumed to target children between ages 5 and 15; community-based PC is assumed to target all individuals of age 2 and above. See Supplementary Information C for visualisations of the associated trends in infection levels.
ᵃFor polygenic inheritance, heritability *h*² = 0.3 means that 30% of the variation in drug efficacy (i.e., the trait) between worms is due to an inherited genetic component. See methods section for details.

to transmission ("exposure heterogeneity") and higher average baseline number of worms per person (Supplementary Information D, Supplementary Figs. D1–2). Here, the baseline average number of worms per person was determined by the transmission rate and the survival rate of environmental life stages (Supplementary Information B, Supplementary Figs. B6 and B7) but not exposure heterogeneity (Supplementary Fig. B8). If elimination was not achieved, transmission conditions did not meaningfully affect the speed at which drug efficacy declined (Supplementary Information D, Supplementary Figs. D3–6). At most, the decline in drug efficacy against *A. lumbricoides* was somewhat slower in less dense worm populations, but only when PC was targeted at school age children.

We performed sensitivity analyses for the impact of baseline allele frequencies in monogenic resistance, degree of heritability in

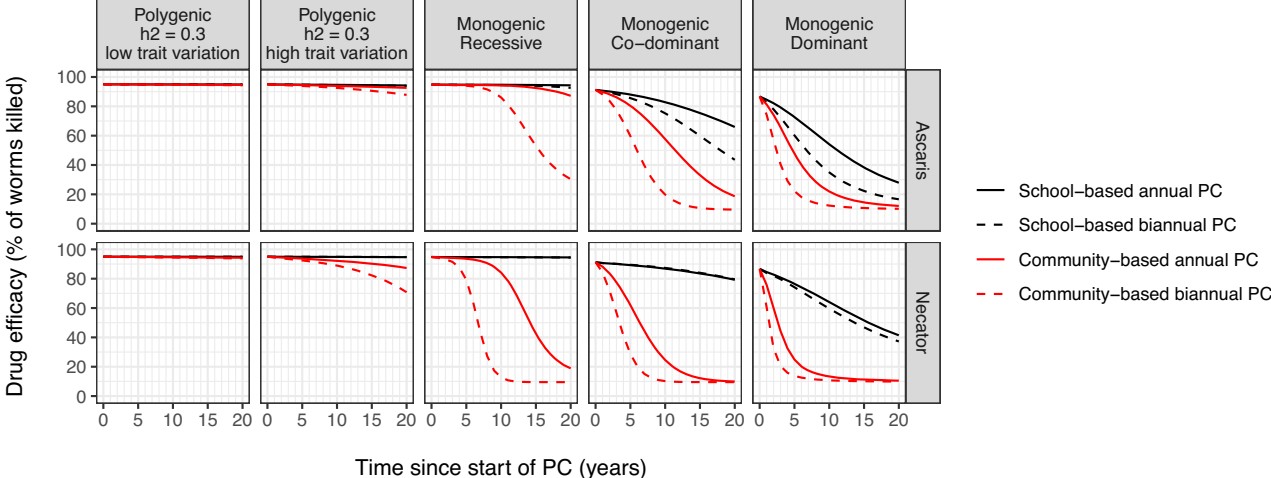

**Fig. 2 | Model-predicted average trends in drug efficacy under different PC strategies and inheritance mechanisms behind parasite drug resistance.** Lines are analogous to the red lines for average drug efficacy in Fig. 1 (third column) and therefore represent settings where elimination was not achieved within 20 years.

Trends are shown for all combinations of PC strategy and frequency (line colours and types), parasite species (rows of panels), and inheritance mechanisms (columns of panels). See Supplementary Information C for detailed results at the level of individual simulations.

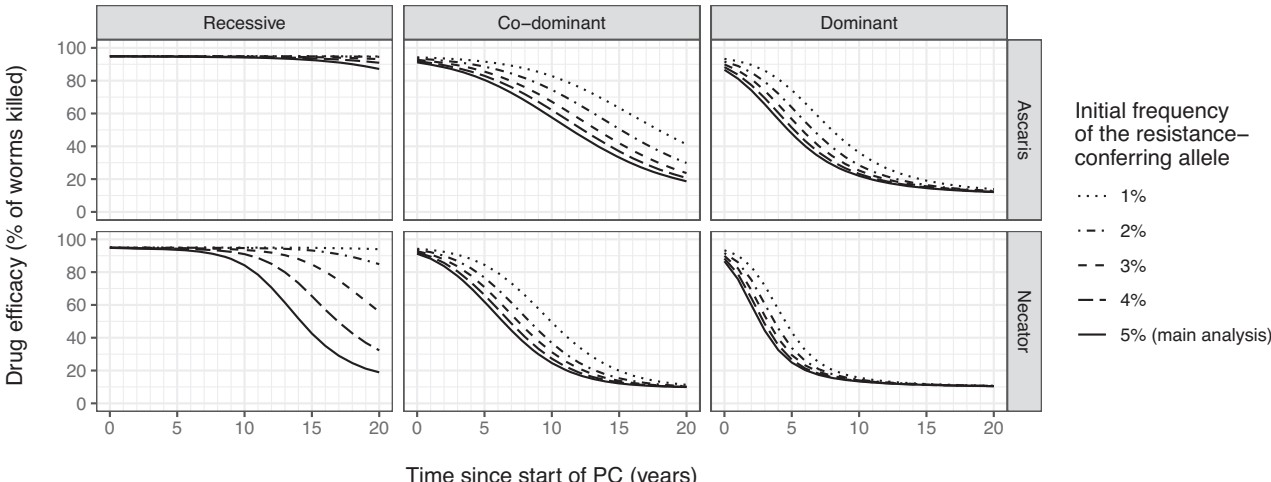

**Fig. 3 | Impact of initial allele frequency on model-predicted trends in drug efficacy during annual community-based PC in case of monogenic drug resistance.** Lines are analogous to the red lines for average drug efficacy in Fig. 1 (third column) and therefore represent settings where elimination was not achieved

within 20 years. See Supplementary Information E, Supplementary Fig. E1 for detailed results at the level of individual simulations and probabilities of elimination.

polygenic resistance, drug efficacy against non-resistant worms, patterns in PC uptake, and the impact of a potential resistance-associated fitness loss. First, lower initial allele frequencies were associated with higher probabilities of elimination and a delayed decline in drug efficacy if elimination was not achieved (Fig. 3). If the initial resistant allele frequency was 1% (instead of 5%), it took longer for drug efficacy to drop below 85%. Depending on the inheritance mechanism, this delay was about 10 ten years (recessive) or as little as approximately 4 (co-dominant) or 2 years (dominant). Second, for polygenic resistance, higher assumed degrees of heritability led to faster declines in drug efficacy if elimination was not achieved, but only marginally affected the probability of elimination itself (Supplementary Information E, Supplementary Figs. E3–4). Third, lower drug efficacy (80%) against non-resistant worms (using monogenic resistance as an example) led to somewhat lower probability of elimination but also somewhat slower declines in drug efficacy if elimination was not achieved (Supplementary Information E, Supplementary Figs. E5 and 6). Fourth,

higher PC coverage was associated with a higher probability of elimination (Supplementary Information E, Supplementary Figs. E7 and 8) but also a faster decline in drug efficacy if elimination was not achieved (Fig. 4). When individual uptake of PC was correlated over time (in contrast to random over time, as in the main analysis), systematically non-treated individuals served as a reservoir of unselected worms, postponing elimination (Supplementary Information E, Supplementary Fig. E8) and slowing down the declining trend in drug efficacy (Fig. 4). These effects were most pronounced for hookworms and settings with lower population coverage of PC. Fifth and last, a fitness loss in terms of lower female worm egg productivity (a 1/3 reduction for full resistance and a 1/6 reduction for partial resistance) resulted in slightly higher probability of elimination and a slower decline in drug efficacy when elimination was not achieved (Supplementary Information E, Supplementary Figs. E9 and 10).

Last, we investigated how drug resistance might manifest as a change in the faecal egg reduction rate (ERR), which is defined as 1

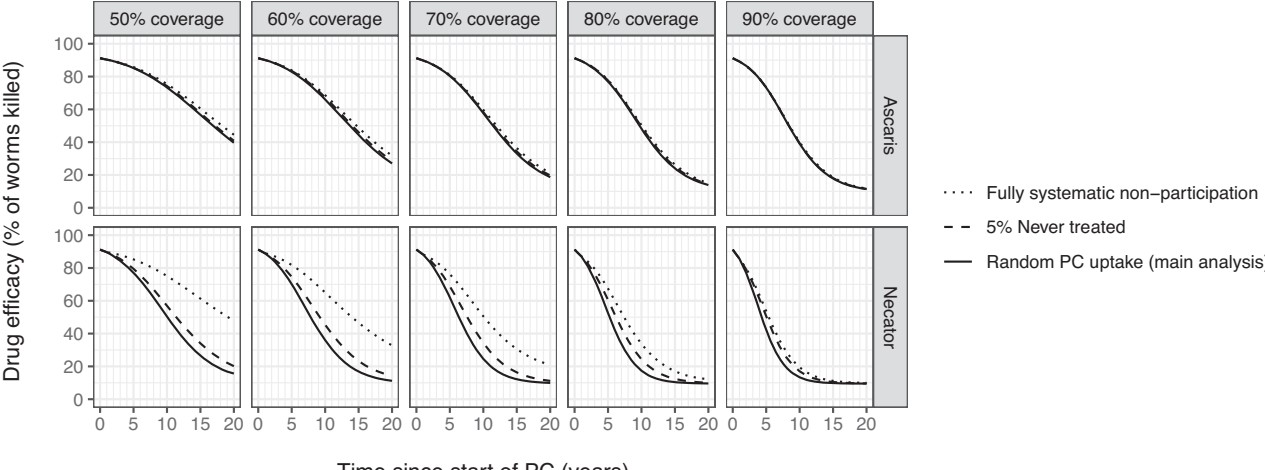

**Fig. 4 | Impact of PC coverage and patterns in individual PC uptake on trends in drug efficacy (co-dominant resistance) during annual community-based PC.** PC coverage is defined in terms of percentage of the targeted population (age 2 and above) that take up PC every round. Random PC uptake means that individual uptake of PC is not correlated over time, such that after 10 rounds >99.9% of individuals that were eligible throughout those 10 years have received at least one treatment. In contrast, "fully systematic non-participation" is the extreme scenario in which always the same individuals are treated such that the proportion never treated is 100% minus the coverage. In between, we define a scenario with "5% never treated" in which after 10 rounds, due to correlation in individual uptake over time, 5% of eligible individuals will have never received a treatment. Lines represent settings where elimination was not achieved within 20 years.

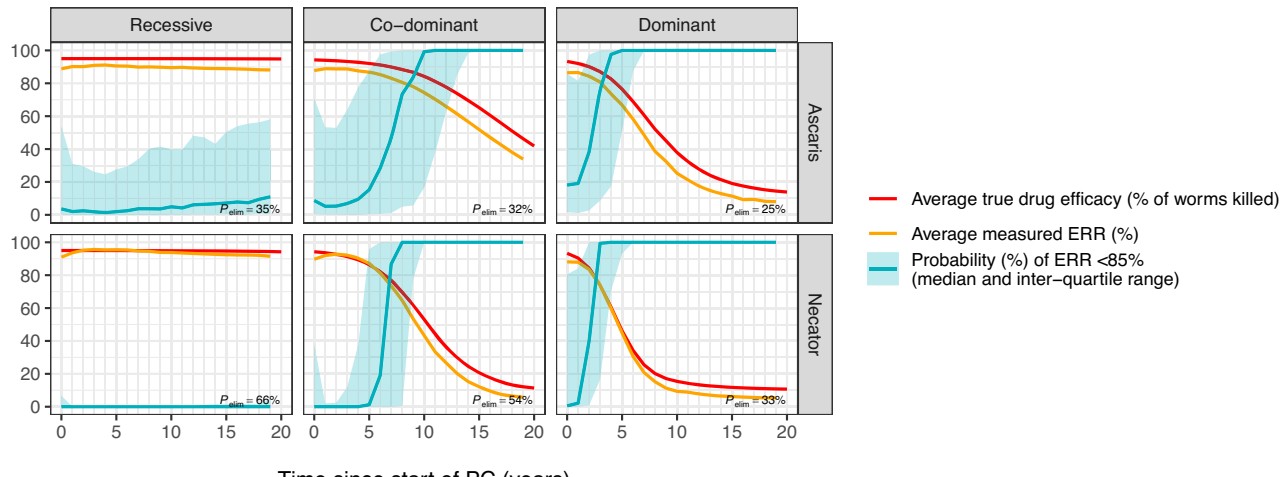

**Fig. 5 | True drug efficacy and measured faecal egg reduction rates (ERR) during annual community-based PC in case of monogenic drug resistance.** For illustrative purposes, the initial frequency of resistant alleles was assumed to be low (1%, as in Fig. 3), such that before the start of PC, the true drug efficacy (red solid line) was very close to 95%. For each panel and time point, the probability of finding an ERR <85% (solid teal line) was based on a 1000 repeated simulated survey of 1000 individuals that were being treated via PC (i.e., all ages of 2 and above). In each simulated survey, individuals were tested with a single Kato-Katz (KK) before treatment and duplicate KK (based on a single faecal sample) two weeks after treatment. ERRs were calculated based on the ratio of pre- and post-treatment arithmetic mean egg counts of all tested individuals. The initial decline in the chance of finding an ERR <85% (teal solid line) is due to the fact that early on during PC programs, there are still relatively many pre-patent worms that are not affected by treatment but who become patent and start producing eggs between the moment of treatment and the post-treatment survey two weeks later. For true (red) and observed drug efficacy (orange), results represent arithmetic means of repeated simulations. As the variability in the probability of finding and ERR <85 was highly skewed, results are represented as medians (teal line) and inter-quartile ranges (teal ribbon).

minus the ratio of the arithmetic mean post-treatment and pre-treatment faecal egg counts in treated individuals. Using a published simulation methodology to determine the optimal survey design for drug efficacy monitoring[31], we simulated 1000 repeated ERR surveys for each time points in each simulation, accounting for variation in repeated egg counts. In each survey, we sampled (with replacement) and tested 1000 individuals before treatment (single Kato-Katz faecal thick smear) and retested the same individuals two weeks after treatment (duplicate Kato-Katz) with the aim to detect reduced efficacy with 90–95% probability[31]. On average, ERR survey results generally followed the "true" drug efficacy in the simulation well, although noise in the egg counting process led to ERR slightly underestimating the true efficacy (Fig. 5). The simulations further showed that it is possible to find ERR <85% even in treatment-naive settings, which is due to the relatively large number of the pre-patent worms, who are not affected by the drug and may become patent and start producing eggs between treatment and follow-up. Still, for settings where the true drug efficacy had started to decline after a few years of PC (notably for co-dominant and dominant inheritance), a measured ERR of <85% was highly indicative of a reduction in true drug efficacy (Supplementary Information

F, Supplementary Fig. F1). In settings with recessive inheritance, a decline in drug efficacy was still relatively rare and could therefore not be so clearly detected with an ERR survey during the simulated 20-year period (but was expected to be detectable with longer duration of PC).

## Discussion

Our study illustrates the dynamic interplay between achieving elimination and emergence of drug resistance in STH, using the first-ever individual-based model for the population dynamics of infection and drug resistance in human helminth infections. Prospects of STH elimination increase when PC is implemented at a higher frequency or targeting a larger part of the population. However, if resistance-conferring alleles are present in a parasite population, more intense PC strategies also increase the speed at which drug efficacy declines. The probability of elimination is lower and risk of resistance spreading is higher when drug resistance is monogenic. This is especially the case with co-dominant or dominant inheritance, in which case we predict drug efficacy to decline noticeably within a decade of PC, even if targeting only school age children. Given that the probability of elimination is lowest in settings with high baseline infection levels and a high degree of parasite aggregation in hosts, if anywhere, drug resistance is likely to emerge there first. This is especially the case if such areas are targeted with more intense PC strategies. Fortunately, under the current WHO-recommended strategy, PC is mostly targeted at school age children such that a large untreated reservoir of parasites remains in adults, slowing down the potential emergence of resistance. Still, even with school-based PC, the risk and speed of resistance emerging is non-negligible, especially for *A. lumbricoides*, which primarily infects children. Given the marked differences in risk and speed of emergence of drug resistance across different genetic mechanisms, there is a need to better understand the main inheritance mechanisms for potential drug resistance in STH and a potential associated fitness loss (although the latter would probably have a modest impact). This will require detailed population-based studies of drug efficacy and parasite genetics. In addition, it would be advisable to survey drug efficacy and population genetics in STH control programs and especially in trial settings where community-wide single-drug treatment against STH is implemented.

Here, we focus on *A. lumbricoides* and the most common hookworm species *N. americanus*. Although *T. trichiura* is also a common STH species, efficacy of benzimidazoles against this species is very low already[19]. However, given the similar age patterns, transmission routes, and parasite life spans for *T. trichiura* and *A. lumbricoides*, if PC was implemented with a drug that is efficacious against trichuriasis, our model predictions would be very similar to those for *A. lumbricoides* presented here. As for the less common and more focally distributed hookworm species *Ancylostoma duodenale*[2], prospects of elimination can be expected to be higher (and risk of resistance lower) than predicted here for *N. americanus* as the number of adult worms underlying observed egg counts is much lower (*A. duodenale* produces two to three-fold more eggs per adult female worm[44]).

Mathematical models allow us to explore and predict the potential repercussions of different PC strategies. In veterinary parasitology, various deterministic[35-40] and stochastic models[41,42] have been developed to investigate anthelminthic drug resistance and potential mitigation strategies. For human helminthiasis, the spread of drug resistance has so far only been studied for lymphatic filariasis, using a deterministic model and assuming that drug resistance is governed by one or at most a few genetic loci[45,46]. Although deterministic models are useful for initial theoretical explorations, they cannot incorporate sufficient detail of multiple interacting real-world heterogeneities, such as individual exposure to infection and PC uptake[45]. A strength of our model is that these heterogeneities are explicitly captured in detail. Also, there is a need for mathematical models to consider the possibility that drug resistance is polygenic, which we do here for the

first time with an individual-based model. We do note that our model only considers a single local community; if instead we would consider multiple communities between which people move, this would make transmission at low levels more stable[47,48] such that the probability of achieving elimination in all connected communities would be lower than we predict here, giving more opportunity for drug resistance to arise.

Based on simulations for school-based and community-based PC, and for varying levels of PC coverage and systematic non-participation of individuals, we clearly show how targeting a higher proportion of the worm population promotes the emergence of drug resistance. Although we did not explicitly investigate the impact of switching from school-based to community-based PC, this concern also applies there (as illustrated in Supplementary Information A, Supplementary Figs. A10 and 11). Refugia of parasites in untreated individuals (and the environment) maintain genetic diversity, preventing fixation of resistance alleles in the parasite population, as observed for schistosomiasis[49]. As such, incomplete population coverage of PC and even systematic non-participation to PC, paradoxically, protect against the emergence of resistance. We further show that higher frequencies of PC promote the emergence of resistance. A such, scaling up the coverage or frequency of PC against STH, which is increasingly considered[10] and trialed[50,51], should be carefully monitored with regard to drug resistance.

For settings where PC does not result in elimination, we predict that the speed at which drug efficacy declines is hardly affected by transmission conditions. This was somewhat surprising as we expected that higher levels of aggregation (i.e., concentration of the parasite population in fewer hosts) would promote the spread of resistance as more worms live in the same host, meaning they can directly produce progeny carrying (combinations of) potentially resistant genes instead of having to first rely on transmission to another host via the environment. However, aggregation turned out to only be important for the probability of elimination; once resistant alleles were sufficiently established in a worm population, the speed at which drug efficacy declined was very similar across different transmission conditions. We further note that we assumed that STH transmission is perennial, whereas in some settings the survival of eggs or larvae in the environment may vary seasonally due to the weather. In settings where PC is implemented in the low transmission season (i.e., when there is little or no environmental refugium for unselected susceptible parasites), we should expect that the selective pressure from PC is higher than we simulate here. As such, if PC is implemented during the low transmission season but does not lead to elimination, this would further promote the spread of resistance as there are fewer susceptible genotypes in circulation.

To prevent or delay the emergence of widespread resistance, we may consider adaptations of PC strategies. For instance, in the control of veterinary parasites, targeted selective treatment of individuals with high infection levels (and presumably highest morbidity levels) is an effective way to maintain refugia of unselected parasites and keep levels of drug resistance manageable[11,42]. However, for human STH, we currently lack the diagnostics or resources to implement such a strategy at a large scale. Another strategy is to perform PC with multiple drugs from different drug families, such as a combination of a benzimidazole and ivermectin, a combination that is already widely used in PC against lymphatic filariasis and that would also be more effective against the STH species *T. trichiura* and *Strongyloides stercoralis*. The extent to which these PC strategies would be able to prevent or delay emergence of resistance is subject to future modelling studies.

Our findings are partially generalisable to other human intestinal helminthiases that are also controlled via PC with a single drug that kills adult worms, such as schistosomiasis. An important difference though is that schistosomes have a longer expected lifespan (~6 years)

and are considered to be more monogamous than STH[52], meaning that the genetic turn-over and mixing of schistosome populations is lower than those of STH (i.e., lower risk and speed of decline in drug efficacy). Also, it is believed that resistance in schistosomiasis against praziquantel is recessive[53], which of all Mendelian inheritance patterns generates the slowest declines in average drug efficacy at the population level. On the other hand, the clonal multiplication of schistosome parasites in the intermediate snail host may also mean that once emerged, drug resistance may actually spread very rapidly. Further modelling studies are required to assess how these contrasting effects balance out.

As for veterinary helminthiases, we already highlighted that resistance is widespread in intestinal worm infections of small ruminants[11,12], cattle[13], and equines[22]. Elimination of veterinary helminthiasis has been rarely achieved and only in combination with education and behaviour change of farmers. The only examples of elimination before resistance arose – that we are aware of – are the nation-wide elimination of echinococcosis in Iceland and New Zealand[54] and the local elimination of *Haemonchus contortus* in a subset of Australian sheep farms, with full-blown drug resistance emerging in all other farms[55].

In conclusion, we confirm that even with the current PC strategy of mainly targeting school age children, drug resistance poses a real threat to the control of STH. If drug resistance is monogenic with co-dominant or dominant inheritance, it may evolve within a decade of PC. This risk is further increased if control programs switch to more intensive PC strategies, which are increasingly considered and trialed. As such, the insights generated by our model are timely and highlight the need for STH control programs to develop and implement strategies to monitor and mitigate the evolution of benzimidazole resistance.

## Methods
### General model structure
For this study we developed a new individual-based model for evolution of drug resistance in STH, building on the existing individual-based WORMSIM model for transmission and control of STH[5,56,57]. WORMSIM predictions for STH have been previously validated against field data on trends in *N. americanus* and *A. lumbricoides* infection levels before and during PC[56,57]. In the new model, basic assumptions about parasite biology and impact of PC are identical to WORMSIM, such that the new model (with drug resistance turned off) reproduced the behaviour of WORMSIM. Here, we purposely focus on *N. americanus* and *A. lumbricoides* as benzimidazoles are not very efficacious against *T. trichiura* (-50% egg reduction rate[19]). The new model was developed as the open-source package "simresist" (version 1.1.1) for R (https://gitlab.com/luccoffeng/simresist). Here we provide a high-level overview of the model; all technical details and differences with previous models are described in Supplementary Information A.

The model simulates the life histories of about 450 individual humans in a local community (the typical geographical scale of transmission) and the life histories of the individual worms living within these humans, using time steps of one week (which is appropriate given that the average lifespan of adult worms is 3 years for *N. americanus* and 1 year for *A. lumbricoides*). Simulated humans are exposed to and contribute to a central reservoir of infection in the environment. Humans contribute infective material to the reservoir as long as they are infected with female worms that produce fertilised eggs, which is only possible after a period of pre-patency (maturation period in the human host of 7–10 weeks) and when at least one male worm is also present in the same host. Due to competition for nutrients and the host immune response, egg production per adult female worm is assumed to decline as the number of female worms in a host increases (i.e., negative density dependence leading to a saturation in total egg output of worms in a single host). The degree of parasite

aggregation within the human population is governed by the level of inter-individual variation in exposure to the central reservoir of infection (by age and random individual factors). Further, the model explicitly simulates individual host participation to PC, accounting for age patterns in uptake and a degree of systematic non-participation. Based on all of the above, the model predicts how drug efficacy translates into the effectiveness of PC to reduce STH infection levels in a human population.

### Genetic mechanisms behind drug resistance
As highlighted in the introduction, for veterinary intestinal helminths, the inheritance mechanism for drug resistance varies between helminth species. Given the paucity of data on inheritance mechanisms in human STH species, we considered two extreme alternative assumptions. The first is that drug resistance is monogenic and autosomal, meaning that it is conferred by a single locus on a non-sex chromosome (e.g., by the beta-tubulin isotype 1 gene) and that it follows a pattern of Mendelian inheritance. The contrasting second assumption is that drug resistance is polygenic, meaning that many loci are involved. Here, we assume that each locus contributes approximately equally to resistance. To avoid the heavy computational burden of explicitly simulating inheritance via many loci, we adopted a quantitative trait model (more details below). Optionally, the concepts of monogenic and polygenic resistance can be mixed in a simulation, although here, we focus on the two extremes of purely monogenic vs. polygenic resistance. Further, based on veterinary literature, we assumed that resistance does not impose a fitness cost[21,22].

Mendelian inheritance was simulated by assigning each worm one of three genotypes: carrying zero, one, or two copies of a resistance-conferring allele. Worm genotype was assigned upon establishment in the host with the probability of a particular genotype proportional to the prevalence of that genotype in the environmental reservoir of infection. Fertilised eggs produced by female worms were randomly assigned genotypes based on the paternal and maternal genotypes available in the same host, according to the principles of Mendelian inheritance[58]. At each time step in the simulation, the distribution of genotypes in the environmental reservoir was updated based on the survival of eggs or larvae in the environmental reservoir and the genotype distribution among excreted eggs. We assumed monogenic resistance was either recessive (only homozygous carriers are fully resistant), co-dominant (heterozygous and homozygous carriers are partially and fully resistant, respectively), or dominant (both heterozygous and homozygous carriers are fully resistant). Based on a previous modelling study for veterinary intestinal helminths[43], we assumed that drug efficacy against fully and partially resistant parasites was reduced by 90% and 40%, respectively, compared to efficacy against non-resistant parasites.

Polygenic resistance was simulated as a quantitative trait of individual worms[58,59], which avoids the computational burden of explicitly simulating all possible combinations of polymorphisms across many loci. The quantitative trait approach is a method to describe phenotypes among offspring as a continuous variable (e.g., like adult height or blood pressure), where an individual worm's trait is assumed to be the sum of an inherited genetic component $x_{gen}$ and a non-inherited component $x_t$ which may vary over time and captures all other remaining sources of variation. If many loci are involved and each locus is assumed to contribute similarly towards the phenotype, the offspring trait can be defined as $x_{trait} = x_{gen} + x_t$ and can be reasonably assumed to follow a normal distribution with expectation equal to the mean of the genetic components of the two parents' trait values ($x_{gen}$), and variation $\sigma_{trait}^2 = \sigma_{gen}^2 + \sigma_t^2$. Here, $\sigma_{gen}^2$ is the variation in trait values among offspring due to genetic recombination and $\sigma_t^2$ governs the temporal phenotypic variation. The heritability $h^2$ of a trait is defined as $\sigma_{gen}^2 / \sigma_{trait}^2$. Here, we assume that $h^2 = 0.3$, based on studies of *Haemonchus contortus* in small ruminants[60]; for illustrative

**Table 2 | Overview of parameter values used in simulations**

| Variable | Values used in simulations |
|---|---|
| Parasite species | *Ascaris lumbricoides* (1-year average lifespan, mostly infecting children) and *Necator americanus* (3-year average lifespan, infecting both children and adults). For both species, individual worm lifespans are assumed to follow a Weibull distribution (see Supplementary Information A for details). |
| **Transmission conditions**[a] | |
| Overall transmission rate (i.e., the annual rate $\zeta$ of exposure and contribution to the environmental reservoir) | Range: 50–250, corresponding to a 10–60% prevalence of infection in the general population, as measured by a single Kato-Katz faecal smear. |
| Exposure heterogeneity (shape parameter $k$) | Range: 0.2–1.0, based on values for level of worm aggregation in literature[61]. |
| Average lifespan of eggs or larvae in environment | Range: half to double the value(s) reported in literature[2,44], i.e., 1–8 weeks for *N. americanus* (literature: 2–4 weeks), and 3 weeks to 3 months for *A. lumbricoides* (literature: 1.5 months). |
| **Preventive chemotherapy (PC)** | |
| Target population and coverage | School age children (ages 5–15) at 95% coverage or community-based (ages 2 and above) at 70% coverage, assuming random individual participation to repeated PC rounds. In sensitivity analyses, we consider a wider range of coverage (50–90%) of community-wide PC and potential systematic non-participation of individuals (5% never treated after 10 rounds). |
| Frequency | Annual or biannual (every 6 months). |
| Average drug efficacy in absence of resistance | 95% of patent worms killed, a plausible value[19,26] chosen for illustrative purposes; non-patent worms are not affected by the drug. In sensitivity analyses, we consider 80%, 90%, and 99%. |
| **Polygenic drug resistance** | |
| Heritability ($h^2$)[b] | 0.3, based on studies of *Haemonchus contortus* in small ruminants[60]; in sensitivity analyses, we consider 0.5 and 0.7. |
| Variation in drug efficacy between worms and treatments | Standard deviation in log-odds of a worm being killed of $\sigma^2_{trait} = 0.178$ ("low" variation, corresponding to a 95%-CI for probability of non-resistant worms being killed of 90.0–97.8%) or $\sigma^2_{trait} = 1.044$ ("high" variation; 95%-CI of 80.0–99.5%). |
| **Monogenic drug resistance** | |
| Initial allele frequency | 5% in Hardy-Weinberg equilibrium (i.e., 90.25% homozygous for the non-resistant allele, 9.5% heterozygous, and 0.25% homozygous for the resistant allele); in sensitivity analyses, we consider initial frequencies of 1%, 2%, 3%, and 4%. |
| Inheritance | Recessive (homozygous carriers are fully resistant), co-dominant (heterozygous and homozygous carriers are partially and fully resistant, respectively), dominant (heterozygous and homozygous carriers are both fully resistant). |
| Phenotype | Drug efficacy: probability of a worm being killed is 95% in the absence of resistance, which drops to 57% (40% reduction) or 9.5% (90% reduction) in the case of partial and full resistance, respectively[43]. |
| **Diagnostic variation in Kato-Katz-based egg counts** | |
| Day-to-day variation in faecal egg density | Gamma variation with shape $k = 0.51$ (*A. lumbricoides*) and $k = 1.0$ (*Necator americanus*)[31]. |
| Slide-to-slide variation in faecal egg density | Poisson variation, assuming stools were properly homogenised[31]. |

[a]Values for population-level transmission parameters were sampled from a uniform distribution on the logarithmic scale.
[b]Heritability ($h^2$) indicates the proportion of variation in drug efficacy among worm offspring that is attributable to genetic factors. If $h^2 = 0$, drug efficacy varies freely between worms and treatments. If $0 < h^2 < 1$, drug efficacy varies between worms but less so between treatments for a given worm due to an inheritable genetic component. If $h^2 = 1$, drug efficacy varies between worms but not between treatments for a given worm because worm susceptibility to the drug is entirely determined by genetics.

purposes we further consider $h^2 = 0.5$ or $h^2 = 0.7$ in sensitivity analyses. Because the quantitative trait $x_{trait}$ can take on any negative or positive value, we let the trait $x_{trait}$ represents the change in log-odds of a worm being killed by the drug relative to the average "wild type" worm with a trait value of zero. For instance, for a worm with trait value $x_{trait} = 0.5$, we define that the odds of being killed by drug treatment are 39% lower ($1 - e^{-x_{trait}}$) lower than a "wild type" worm with trait $x_{trait} = 0$. Thus, the actual probability of each worm being killed by drug treatment depends on their trait and the user-defined probability for a worm with trait $x_{trait} = 0$ to be killed. In our simulations, we considered two levels of trait variation $\sigma^2_{trait}$, such that in a worm population with a mean trait value of 0, the lower bound of a 95%-confidence interval for the probability of a worm being killed was 5 ("low" variation) or 10 percentage points ("high variation") under the average drug efficacy (see Table 2 for details).

**Simulations**
First, for each parasite species, we created a database of 1000 random baseline (pre-PC) population states (400–440 humans, reflecting the size of an average endemic community) in a range of random transmission conditions. Transmission conditions were defined in terms of

three parameters. The first was the average rate at which humans are exposed and contribute to the environmental reservoir of infection (i.e., the overall transmission rate), which was assigned a range of values (Table 2) such that the majority of simulations represented settings with a baseline prevalence of infection of 10–60% in the general population, as measured by a single Kato-Katz faecal smear (Supplementary Information B, Supplementary Fig. B5). The second parameter was the level of inter-individual variation in exposure to the environmental reservoir (i.e., the shape parameter of the gamma distribution for exposure heterogeneity), which was allowed to vary between 0.2 and 1.0, based on a previous review of literature[61]. The third transmission-related parameter was the suitability of the environment for survival of eggs or larvae (i.e., the average lifespan of eggs or larvae), which was allowed to vary between half and twice the values reported in literature (average of 2–4 weeks for *N. americanus*, 1.5 months for *A. lumbricoides*[2,44]) to capture potential geographical variation. Per simulation, each of the three parameter values for transmission conditions was drawn from a uniform distribution on the logarithmic scale. Baseline population states were generated by warming up human demography for 250 years in one-year time steps. Next, worm establishment was warmed up (alongside human

demography) for another 50 years in one-week time steps. If after this 300-year warm-up no worms were present in the simulation, a new random set of transmission parameters was drawn and the process was repeated until 1000 baseline states with at least 1 worm had been created.

Next, for each baseline state, we simulated the impact of various PC scenarios (Table 2) over the course of 20 years in one-week time steps. PC scenarios were defined in terms of the PC target population and coverage (school-aged children, ages 5–15 at 95% coverage, or community-based, ages ≥2 at 70% coverage), PC frequency (once or twice per year), "normal" drug efficacy (95% of worms killed, which based on observed egg reduction rates is a plausible value[19,26] and was chosen for illustrative purposes), and the assumed genetic mechanisms behind drug resistance. Parasite population genetics were assumed to be in equilibrium until right before the start of PC. That is, the initial distribution of quantitative traits was assumed to follow a normal distribution with mean zero and variance $\sigma^2_{trait}$ (either low or high variation, as described above). For monogenic drug resistance, initial genotype frequencies were generated assuming an allele frequency of 5% in Hardy-Weinberg equilibrium. See Table 2 for an overview of all quantifications of model parameters for transmission, PC, and drug resistance.

### Reporting summary
Further information on research design is available in the Nature Portfolio Reporting Summary linked to this article.

## Data availability
The model-simulated data that support the findings of this study are available at https://doi.org/10.6084/m9.figshare.24521431. The code to generate and process it the simulated data are available at https://gitlab.com/luccoffeng/simresist-nature-comms-sth-paper.

## Code availability
The source code of the individual-based model (including extensive documentation) is publicly available at https://gitlab.com/luccoffeng/simresist. A frozen version of the source code that was used for this paper (v1.1.1) is available at https://doi.org/10.5281/zenodo.10442358.

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

## Acknowledgements

The authors gratefully acknowledge the WHO/TDR Special Programme for Research and Training in Tropical Diseases, and in particular Dr. Annette Kuesel for their support of the development of a prototype of the drug resistance simulation model, which was developed for onchocerciasis and soil-transmitted helminths as part of a grant from WHO/TDR Special Programme for Research and Training in Tropical Diseases to Erasmus MC Rotterdam (project ID B40126, SJdV and WAS). In addition, this work was supported by a grant from the Dutch Research Council (NWO, grant 016.Veni.178.023, LEC). LEC is further grateful to Prof. Dr. Warwick Grant, Prof. Dr. Jon Gilleard, Dr. Stephen Doyle, Dr. Kathryn Kemper, Prof. Dr. Ray Kaplan, Dr. Dave Leathwick, Dr. Robert Dobson, and Prof. Dr. Bruno Levecke for letting him pick their brain about the genetics of drug resistance in human and veterinary parasites.

## Author contributions

L.E.C. conceived of the study, designed and programmed the simulations model, performed the simulations and analysis, visualised and

interpreted the study results, and drafted the manuscript and its supplements. W.A.S. conceived of the study, interpreted the study results, and contributed to the revision of the manuscript. S.J.d.V. conceived of the study, interpreted the study results, and contributed to the revision of the manuscript.

## Competing interests

The authors declare no competing interests.
