## [Peer Review File · Nature Communications]

Predicting the risk and speed of drug resistance emerging in soil-transmitted helminths during preventive chemotherapyREVIEWER COMMENTS

Reviewer #1 (Remarks to the Author):

This modelling study by Coffeng and colleagues investigates the potential emergence of anthelmintic resistance in roundworms (*Ascaris* and hookworm) resulting from mass treatment. It is unusual for a sophisticated parasite transmission model to incorporate genetic information and so their approach is novel and has the potential to generate insightful findings. The mechanisms of genetic resistance have been thoughtfully considered (perhaps due to picking the brains of so many worm geneticists).

While some results are of interest, such as the impact of the type of resistance mutation (monogenic dominant vs recessive vs polygenic) and different resistance dynamics between the two nematode species, the key findings the authors have chosen to focus on are largely unsurprising - resistance emerges more rapidly given more frequent treatment and coverage of the whole population (compared with treating school children only). For publication in *Nature Comms*, I would suggest adding additional analyses from their model, which would make the results more policy relevant and increase the impact of this work:

1. It would be valuable to relate changes in treatment efficacy (killing of adult worms) to observable diagnostic outcomes, such as the "egg reduction rate" (ERR) and "cure rate" (CR) used in most field studies and advised by WHO. Given the expected egg output per worm, at what point do changes in ERR or CR indicate the emergence of resistance, versus natural variation in egg output (or reinfection)? This question should be dealt with as a further simulation and not just additional discussion. Results from this analysis could lead to a more evidence-based thresholds for ERR or CR.

2. Following from this, what do the current range of observed ERR and CR from STH field surveys indicate about ongoing selection from mass treatment? E.g. are current reported values consistent with no selection? Again this should be estimated probabilistically, not just added to the discussion.

3 How does the probability of elimination vs resistance change with varying levels of treatment coverage (other than 70% for whole-population)? Is there an optimal population coverage for achieving elimination while avoiding resistance?

4 How sensitive are the probabilities of elimination / resistance to the proportion of people "never treated"? E.g. how do the results vary for 70% annual coverage where this is randomly resampled from the entire population, compared to scenarios where 30% remain fully untreated? (Update - after reading the supplementary a parameter for systematic non-adherence is included in the model, however failing to explore how variation in this parameter affects model output is a missed opportunity)

Further suggested changes:

5. As the analysis can be chopped up in so many ways this creates the potential for an unmanageable number of scenarios (two parasites species, five models of resistance, four types of treatment coverage etc) I would therefore suggest restricting the focus onto what is most biologically and epidemiologically plausible. A single figure (e.g. the current Fig 2) could discuss the differences between monogenic and polygenic resistance. While there may be disagreement on the most likely mechanism, all the examples of parasite resistance that I'm aware of are monogenic and so these scenarios should arguably be the focus for the remainder of the study. Similarly, mass treatment more than once annually is unusual and difficult for control programs to maintain, therefore assuming annual treatment for the majority of scenarios seems most appropriate.

6. The title and introduction, which focus on "soil-transmitted helminths" jars with the analysis which is only on two nematode species. Extend the analysis to include whipworm or change the title. Can the "*Necator americanus*" model be generalised to "hookworm"?

7. The abstract is too casual for a high impact publication (e.g. "We did this for twenty years...") and fails to present noteworthy results. Consider revising.
8. The term "preventative chemotherapy" is misleading. Anthelmintic treatment is not given prophylactically. Consider replacing this term with "mass deworming", "population-wide deworming / school-based deworming", or "mass treatment" throughout.
9. Fig 1. "Average female worm load" is ambiguous. I presume this means the "average load of female worm", but it could also mean the "average load of worms in [human] females".
10. Fig 1. Given the uncertainty in the model and parameters, the probability of elimination can be presented without a decimal, e.g 65% and 38% are appropriate degrees of accuracy.
11. It's surprising that the worm burden doesn't fluctuate annually after treatment - is the effect of mass treatment smoothed over the population? (particularly surprising given a population size of 450)
12. The term "semi-annual" is ambiguous. For twice annual treatment, use "biannual". As mentioned above, I would limit the use of biannual scenarios as this is not typically employed in endemic countries over periods as long as twenty years.
13. Figure 4 is not particularly informative - consider replacing with the novel analysis suggested above.
- 14a. Table 2 contains the only substantive descriptions of the model in the main text and should be improved.
- 14b. Sampling transmission parameters from a uniform distribution is unusual - typically a gamma distribution is used, where the shape parameter gives the aggregation ("k" here).
- 14c. Usually "k" refers to the adult worm aggregation rather than the transmission aggregation - are the two values equivalent?
- 14d. I don't understand the transmission rate from the table description - "Range: 50–250, corresponding to a 10% to 60% prevalence of infection as measured by a single Kato-Katz faecal smear." The transmission rate, or force of infection, is normally a rate of adult worm acquisition per unit time. Inspecting the supplementary text, it appears this is
- 14e. The total population size is only described in the supplementary (~450 people at equilibrium), this should be included in the main text and justified. I would have expected a larger population to be modelled, or include some migration into the population.
- 14f. Clarify that the worm lifespan follows a Weibull distribution.
- 14g. "Variation in drug efficacy between worms and treatments (95%-confidence interval (CI) of drug efficacy and corresponding σ on the log-odds scale)" - I don't understand this text and these values. Can they be presented on an absolute scale?
- 14h. "90% reduction in drug efficacy in case of full resistance, and 40% reduction in drug efficacy in case of partial resistance [43]. This corresponds to 9.5% and 57% probability of a parasite being killed, respectively, compared to a 95% probability in case of absence of resistance." - this description is cumbersome. Consider simplifying to "Drug efficacy. Probability of a worm being killed is 95% in the absence of resistance, which drops to 57% or 9.5% in the case of partial and full resistance, respectively".
15. In general, more technical details from the supplementary should be included in the main text. It should be possible to understand the key features of the model from reading the main text alone.

16. The discussion is lengthy relative to the results section and could be condensed.

17. Data availability - the authors should make the "model generated data" available from a repository and it should be possible to reproduce the figures from the code and data. There is no reason to restrict access to synthetic data.

Reviewer #2 (Remarks to the Author):

The manuscript is well-structured and well-penned, offering significant insights into the parameterization of hookworm (*Necator americanus*) and *Ascaris*, drawn meticulously from extensive literature on their biology and epidemiology.

The authors have proficiently incorporated means to represent varied resistance genotypes encompassing both singular and multiple genes within the simulation.

The simulation harbors a profound public health message, emphasizing the substantial risk and rapid evolution of benzimidazole resistance in Soil-Transmitted Helminths (STH), thereby underscoring the immediate need for the development of vigilant surveillance and responsive mitigation strategies.

However, there are critical elements in the model that necessitate addressal, hindering its immediate acceptance for publication.

1. Lack of Model Validations:

The manuscript lacks validations for the newly introduced model. Even though this model builds upon previous works on worm infection simulation, it demands re-validation against clinical aspects of the diseases. The validations may include the following points but not limit to:

- Worm distribution's variability with age and transmission intensity and supporting data for the same. For instance, Figure B1 suggests a worm population size of over 10,000 worms, approximating to 25 worms per individual. Validation data for this is imperative.
- Corroboration of the age-stratified prevalence at equilibrium with existing data or specific demographics. How the shape changing on different transmission intensity.
- Examination of the relationship between prevalence and the transmission rate.
- Validation of the prevalence of egg presence in feces.

2. Population Size:

A human population size of 400 appears insufficient to accurately simulate drug resistance in an individual-based simulation. Especially considering a 5-10% prevalence and 40% of the population aged 0-15, the resultant 16 individuals harboring parasites provide a limited view of the system dynamics when PC targets only these individuals.

3. Role of Immunity:

Immunity is pivotal in transmission dynamics and, presumably being age-dependent, could potentially influence the infection duration and the contribution to the Force of Infection (FOI) in adults despite equivalent environmental exposure levels.

4. Cost of Resistance:

There is a prevailing hypothesis of an inherent cost to resistance, potentially impacting the number of eggs per generation or the lifespan of the resistance genotype. Any model addressing drug resistance should incorporate, at minimum, a foundational form and sensitivity analysis reflecting the potential impacts of these resistance costs.

5. Drug Impact on Eggs:

The drugs utilized in Mass Drug Administration (MDA) serve not only to exterminate the parasites but also to alter the egg-positive rate, both directly and indirectly as delineated in the literature.

Thus, validation regarding the egg-positive rates pre and post-MDA is vital for an accurate drug model.

Minor Concerns:

- The warm-up period of 300 years appears excessive; a justification for this duration is needed.
- An explanation for opting for a weekly timestep over a daily one for the simulation is required.
- Clarification is needed on the reproductive capabilities of a male worm during multiple infections, specifically whether a long-infected male worm can still reproduce by mating with a newly infected female one.

While the manuscript is commendable for its organization, clarity, and significant public health implications, addressing the highlighted concerns is essential to enhancing its overall validity and relevance. The resolution of issues related to model validation, consideration of immunity, population size, and cost of resistance will considerably strengthen the credibility and contribution of this work to the scientific community.

Tran Dang Nguyen
Research Assistant Professor
The Pennsylvania State University

Reviewer #3 (Remarks to the Author):

This work is of genuine importance, for as the authors point out, little has been done to date examining the risk of emerging resistance on the control of STH infections globally, or within endemic communities. The absence of standardized or easily applied methods to monitor for genetically mediated resistance in human helminth infections raises concerns about the ability to detect relevant reductions in treatment response, especially in the setting of targeted chemotherapy or mass drug administration programs.

That said, there are some aspects of the manuscript that would benefit from greater clarity and potentially analysis.

1. The authors estimate a baseline "efficacy" of 95% for albendazole against both *Ascaris* and *Necator* (Fig 1, Fig 4). It's not clear if that refers to cure rate (CR) or Egg Reduction Rate (ERR), which are two useful but clearly distinct metrics for measuring drug response.
2. It is well recognized that the benzimidazole drugs do not have the same activity against the three major STHs (*Ascaris*, *Trichuris*, hookworms), and that there are intrinsic differences that should be taken into account. Given that cure rates for *Ascaris* are typically much higher than those reported for hookworm, accounting for the difference might impact the models presented. The authors should comment on whether these have been considered.
3. It is worth distinguishing between "efficacy" and "effectiveness," the latter being how well the drug works in regular practice, as opposed to how it works in the setting of a controlled trial.
4. The authors reference studies showing host factors that may alter response to treatment, yet it's not clear from the analysis that this was accounted for.
5. The authors use a baseline prevalence of approximately 60% as a starting point for the model. However, this is a very high prevalence and probably not the case in most endemic communities today, since over the past 2 decades endemic communities are frequently characterized by low prevalence (20-30%) and low intensity infections. It would be helpful if the authors commented on how the baseline prevalence chosen might influence the outcome of the predictive modeling they have carried out.

Response to reviewer comments

We thank the reviewers for their positive and constructive comments. Overall, the reviewers agree on the importance and relevance of our findings, but also suggest some additional analyses. We have therefore performed additional simulations for (1) the impact of PC coverage and patterns in systematic (non-)participation to PC, (2) drug efficacy as measured in a survey based on faecal egg counts with Kato-Katz thick smears, and (3) the impact of a potential fitness loss associated with resistance.

Below, we address each reviewer comment in detail, where we list our responses in grey boxes, like here. In the package of resubmitted manuscript files, we also include a copy of the manuscript with all changes in tracked changes.

Reviewer #1

This modelling study by Coffeng and colleagues investigates the potential emergence of anthelmintic resistance in roundworms (*Ascaris* and hookworm) resulting from mass treatment. It is unusual for a sophisticated parasite transmission model to incorporate genetic information and so their approach is novel and has the potential to generate insightful findings. The mechanisms of genetic resistance have been thoughtfully considered (perhaps due to picking the brains of so many worm geneticists).

While some results are of interest, such as the impact of the type of resistance mutation (monogenic dominant vs recessive vs polygenic) and different resistance dynamics between the two nematode species, the key findings the authors have chosen to focus on are largely unsurprising - resistance emerges more rapidly given more frequent treatment and coverage of the whole population (compared with treating school children only). For publication in Nature Comms, I would suggest adding additional analyses from their model, which would make the results more policy relevant and increase the impact of this work:

1. It would be valuable to relate changes in treatment efficacy (killing of adult worms) to observable diagnostic outcomes, such as the "egg reduction rate" (ERR) and "cure rate" (CR) used in most field studies and advised by WHO. Given the expected egg output per worm, at what point do changes in ERR or CR indicate the emergence of resistance, versus natural variation in egg output (or reinfection)? This question should be dealt with as a further simulation and not just additional discussion. Results from this analysis could lead to a more evidence-based thresholds for ERR or CR.

Response 1

Thank you for this suggestion. We have now performed additional analyses for model-predicted egg reduction rates (ERR), based on a survey design using Kato-Katz (KK), optimised to detect reduced efficacy with 90-95% power (1,000 individuals per survey, single KK pre-treatment, duplicate KK post-treatment; [1]). For every PC round in each simulation, we simulated 1,000 repeated surveys to calculate the probability of finding a ERR <85%, which was taken to be indicative of reduced drug efficacy. We did not calculate cure rates (CR) as they should not be used in field studies for drug efficacy monitoring (as per WHO guidelines) given there are major confounding factors such as pre-treatment infection intensity [2]; CRs are mainly useful for randomized controlled trials, in which confounding factors can be better controlled.

Based on the new simulations for ERR surveys, we have added a new figure (Fig 5) for trends in true drug efficacy and the probability of measuring ERR <85% over time (below). From these predicted trends, we calculated the predictive value of an ERR <85% for an average level of true drug efficacy <90% in the worm population (i.e., 5 percentage points below the “normal” efficacy against non-resistant worms), as well as its predictive value for not achieving elimination within 20 years. The initial decline in the chance of finding an ERR <85% is due to the fact that early on during PC programs, there are still relatively many pre-patent worms (who are not affected by treatment) who become patent and start producing eggs between the moment of treatment and the post-treatment survey two weeks later.

To present this figure, we have also added a paragraph to the end of the results section.

2. Following from this, what do the current range of observed ERR and CR from STH field surveys indicate about ongoing selection from mass treatment? E.g. are current reported values consistent with no selection? Again this should be estimated probabilistically, not just added to the discussion.

Response 2

We agree that such an analysis is highly needed. However, drug efficacy against STH is not systematically monitored, and as mentioned in the introduction, existing survey data are based on a WHO-recommended survey design that results in biased estimates (overestimation) of drug efficacy: “Detecting true drug resistance is further complicated by the fact that the current WHO-recommended survey design for evaluating drug efficacy [3], which is based on selection of egg-positive individuals before treatment, leads to overestimation of drug efficacy [1,4].” As such, there are no reliable data for such an analysis. For exactly this reason, based on earlier work [1,4], the first author is currently helping WHO rewrite its technical manual for drug efficacy monitoring in STH and schistosomiasis control programs.

3 How does the probability of elimination vs resistance change with varying levels of treatment coverage (other than 70% for whole-population)? Is there an optimal population coverage for achieving elimination while avoiding resistance?

Response 3

Thank you for this suggestion. We have now performed additional simulations and present these as sensitivity analyses (Figure 4 + new text in the results section, and additional Figures E7-8 in Appendix E). In short, there is no sweet spot; higher coverage means

somewhat higher prospects of elimination (although the overall probability of elimination never exceeds 45%) but also a faster decline in drug efficacy in settings where elimination is not achieved (and vice versa). As such, we have only added the following bold text to the discussion section:

*“Based on simulations for school-based and community-based PC, **and for varying levels of PC coverage and systematic non-participation of individuals**, we clearly show how targeting a higher proportion of the worm population promotes the emergence of drug resistance.”*

With these additional analyses for the impact of PC coverage, our point in the discussion about prospects of elimination remains the same:

“As such, incomplete population coverage of PC and even systematic non-participation to PC, paradoxically, protect against the emergence of resistance (but also decrease the prospects of elimination) [5].”

4 How sensitive are the probabilities of elimination / resistance to the proportion of people "never treated"? E.g. how do the results vary for 70% annual coverage where this is randomly resampled from the entire population, compared to scenarios where 30% remain fully untreated? (Update - after reading the supplementary a parameter for systematic non-adherence is included in the model, however failing to explore how variation in this parameter affects model output is a missed opportunity)

Response 4

Thank you for highlighting that this might not be directly obvious or that the material in the discussion is too brief to do justice to this important aspect. We have now performed additional simulation for the impact of systematic non-participation, which are summarized in Appendix E, Figures E7-8 (along with the impact of coverage itself, see Response 3).

In the results section we have added:

*“Fourth and last, higher PC coverage was associated with a higher probability of elimination (Appendix E, Figures E7-8) but also a faster decline in drug efficacy if elimination was not achieved (**Error! Reference source not found. 4**). When individual uptake of PC was correlated over time (in contrast to random over time, as in the main analysis), systematically non-treated individuals served as a reservoir of unselected worms, postponing elimination (Appendix E, Figure E8) and slowing down the declining trend in drug efficacy (**Error! Reference source not found. 4**). These effects were most pronounced for hookworms and settings with lower population coverage of PC.”*

In the discussion section we have added:

*“Based on simulations for school-based and community-based PC, **and for varying levels of PC coverage and systematic non-participation of individuals**, we clearly show how targeting a higher proportion of the worm population promotes the emergence of drug resistance.”*

Further suggested changes:

5. As the analysis can be chopped up in so many ways this creates the potential for an unmanageable number of scenarios (two parasites species, five models of resistance, four

types of treatment coverage etc) I would therefore suggest restricting the focus onto what is most biologically and epidemiologically plausible. A single figure (e.g. the current Fig 2) could discuss the differences between monogenic and polygenic resistance. While there may be disagreement on the most likely mechanism, all the examples of parasite resistance that I'm aware of are monogenic and so these scenarios should arguably be the focus for the remainder of the study. Similarly, mass treatment more than once annually is unusual and difficult for control programs to maintain, therefore assuming annual treatment for the majority of scenarios seems most appropriate.

Response 5

We assume that with this remark, the reviewer is referring to the additional simulation scenarios that would be (and that we have) created to address their previous comments about the impact of coverage and patterns in individual uptake of PC. We agree that it makes most sense to focus these additional analyses on monogenic resistance and annual treatment. In addition, we have focused on community-wide PC to also explore and illustrate a potential impact on probability of elimination (which were already very low for school-based PC scenarios).

6. The title and introduction, which focus on "soil-transmitted helminths" jars with the analysis which is only on two nematode species. Extend the analysis to include whipworm or change the title. Can the "*Necator americanus*" model be generalised to "hookworm"?

Response 6

As indicated in the first paragraph of the methods section, we purposely focus on *N. americanus* (it said "hookworm" before revision) and *A. lumbricoides* as benzimidazoles are not very efficacious against *T. trichiura*. In the discussion, we have added material to indicate how our findings apply to *T. trichiura* and the other major hookworm species *Ancylostoma duodenale*:

"Here, we focus on A. lumbricoides and the most common hookworm species N. americanus. Although T. trichiura is also a common STH species, efficacy of benzimidazoles against this species is very low already [6]. However, given the similar age patterns, transmission routes, and parasite life spans for T. trichiura and A. lumbricoides, if PC was implemented with a drug that is efficacious against trichuriasis, our model predictions would be very similar to those for A. lumbricoides presented here. As for the less common and more focally distributed hookworm species Ancylostoma duodenale [7], prospects of elimination can be expected to be higher (and risk of resistance lower) than predicted here for N. americanus as the number of adult worms underlying observed egg prevalence is much lower (A. duodenale produces two to three-fold more eggs per adult female worm [8])."

With these additions, we hope that the reviewer and editor agree it is justified to keep the title as it is.

7. The abstract is too casual for a high impact publication (e.g. "We did this for twenty years...") and fails to present noteworthy results. Consider revising.

Response 7

We have revised the abstract to be less casual and to provide more noteworthy results. We have moved the most important (summary of) results from the discussion to the results

section, and now provide more context for the study's implications in the discussion section (additions in bold, removed parts stricken out):

*"METHODS: Using a newly developed individual-based model, we predict the population dynamics of human STH infections and drug resistance during **20 years of regular preventative chemotherapy (PC)**. ~~We do this for a 20-year time period, considering different PC strategies and genetic mechanisms behind resistance.~~ **We compare different PC strategies (school versus community-based deworming) and a range of potential genetic mechanisms behind resistance (monogenic vs. polygenic).**"*

*RESULTS: **With the current PC strategy of mainly targeting children in schools, drug resistance may evolve in STH within a decade.** More intense PC strategies increase the prospects of STH elimination, but also increase ~~selective pressure and thus~~ the speed at which drug efficacy declines, especially when implementing community-based PC (population-wide deworming). ~~instead of school-based PC (children only).~~ **The risk of resistance is highest when resistance is monogenic, especially with co-dominant or dominant inheritance.***

*DISCUSSION: ~~Even with the current PC strategy of mainly targeting children in schools, drug resistance may evolve within a decade. More intensive PC strategies would even further increase the risk and speed of resistance emerging. As such, our findings are timely and~~ **Our predictions confirm that the concerns regarding drug resistance in human STH are founded. Although in some areas, PC against STH has been implemented for about a decade, drug efficacy has not been structurally monitored, or incorrectly so. These findings highlight the need to develop and implement strategies to monitor and mitigate the evolution of benzimidazole resistance.**"*

8. The term "preventative chemotherapy" is misleading. Anthelmintic treatment is not given prophylactically. Consider replacing this term with "mass deworming", "population-wide deworming / school-based deworming", or "mass treatment" throughout.

Response 8

We understand why the reviewer might raise this point, but choose to leave this terminology unchanged. "Preventive chemotherapy" is the official WHO terminology used for this type of intervention (<https://iris.who.int/handle/10665/258983>) and is used across a range of neglected tropical diseases, including STH. The "preventive" or "prophylactic" aspect of it is that drug treatment is administered at the population level, without individual diagnosis, with the aim to reduce and prevent further accumulation of the disease burden in the population. To avoid similar concerns among readers, we now further clarify in the introduction:

*"In the short term, control of STH relies heavily on ~~preventive chemotherapy (PC): regular large-scale deworming of high-risk groups with mebendazole or albendazole, which are both benzimidazole derivatives [9].~~ **Deworming is implemented as preventive chemotherapy (PC), meaning that all treatment-eligible individuals in endemic populations are offered treatment, without individual diagnosis.**"*

9. Fig 1. "Average female worm load" is ambiguous. I presume this means the "average load of female worm", but it could also mean the "average load of worms in [human] females".

Response 9

Yes, this is the average load of female worms, as explained in the figure caption (*“the average number of female worms per person”*). To keep the figure clean-looking (as well as the supplemental figures), we prefer to keep the labels within the figure itself as they currently are, limited to one line.

10. Fig 1. Given the uncertainty in the model and parameters, the probability of elimination can be presented without a decimal, e.g. 65% and 38% are appropriate degrees of accuracy.

Response 10

Thank you for this suggestion. We have reduced the number of decimals in all figures (including Appendices) and Table 1.

11. It's surprising that the worm burden doesn't fluctuate annually after treatment - is the effect of mass treatment smoothed over the population? (particularly surprising given a population size of 450).

Response 11

The reviewer rightly points out that infection levels should fluctuate between PC rounds. As a matter of fact, in our simulations, infection levels do actually drop and bounce back between PC rounds (as illustrated in Appendix A). However, we consciously chose to not show these fluctuations between the annual treatment rounds, as they would make the figures very hard to read. Rather, we only plot one data point per simulated year (just before the implementation of a PC round) such that the trends lines represent (approximately) the peak infection level in each year. We now clarify this in the caption to figure 1:

“Thin lines represent single simulations (N = 1,000), where each line is based on one simulated data point per year, timed right before a PC round; for an example of how infection levels fluctuate between PC rounds, see Appendix A. A grey line colour indicates”

12. The term "semi-annual" is ambiguous. For twice annual treatment, use "biannual". As mentioned above, I would limit the use of biannual scenarios as this is not typically employed in endemic countries over periods as long as twenty years.

Response 12

Throughout the manuscript, figures, appendices, and software code, we have replaced the term “semi-annual” with “biannual”. We have also double-checked that this term is used by WHO to mean “twice per year” (<https://iris.who.int/handle/10665/258983>) and not “every two years”.

13. Figure 4 is not particularly informative - consider replacing with the novel analysis suggested above.

Response 13

To make space for figures on detectability of resistance through drug efficacy surveys (Response 1) and the impact of coverage and patterns in individual PC uptake (Response 3 and Response 4), we have moved Figure 4 to Appendix E (sensitivity analyses).

14a. Table 2 contains the only substantive descriptions of the model in the main text and should be improved.

Response 14

See Response 22.

14b. Sampling transmission parameters from a uniform distribution is unusual - typically a gamma distribution is used, where the shape parameter gives the aggregation ("k" here).

Response 15

The uniform distribution that the reviewer is referring to, is used to let transmission conditions vary between repeated simulations and to achieve a range of plausible and relevant baseline infection levels. Within each simulation, though, we do use a gamma distribution to sample inter-individual variation in contribution and exposure to transmission, which indeed results in a skewed distribution of worms across human hosts (with aggregation "k"). To make this clear, we have adapted the following text in the methods section:

"All Per simulation, each of the three parameter values for transmission conditions were was drawn from a uniform distributions on the logarithmic scale."

as well as the foot note to Table 2:

"Parameter values related to transmission Values for population-level transmission parameters were sampled from a uniform distribution on the logarithmic scale."

14c. Usually "k" refers to the adult worm aggregation rather than the transmission aggregation - are the two values equivalent?

Response 16

Yes, in an equilibrium situation they are the same. Analogous to the distribution of transition times in heterogeneous Poisson processes (<https://www.jstor.org/stable/2632769>), when events, such as worm acquisition, occur to individuals following a Poisson process with inter-individual heterogeneity following a gamma distribution with shape k , the total number of events (worms) per person in a pre-defined time window (e.g., the lifespan of the worm) follows a negative binomial distribution with shape k ([https://doi.org/10.1016/0022-2496\(83\)90043-3](https://doi.org/10.1016/0022-2496(83)90043-3)). In a non-equilibrium situation, the effective aggregation of the worm population may (temporarily) be higher due to additional heterogeneity caused by, e.g., incomplete coverage of PC and inter-individual variation in treatment effects (which we explicitly captured in our individual-based model). To clarify, we have adapted the description in Table 2 to: ***"Range: 0.2–1.0, based on values reported for level of worm aggregation in literature [10]."***

14d. I don't understand the transmission rate from the table description - "Range: 50–250,

corresponding to a 10% to 60% prevalence of infection as measured by a single Kato-Katz faecal smear." The transmission rate, or force of infection, is normally a rate of adult worm acquisition per unit time. Inspecting the supplementary text, it appears this is

Response 17

It seems the reviewer's comment was truncated ("... it appears this is [...]"); we assume that the reviewer's point pertains to the fact that the term "transmission rate" actually stands for "exposure and contribution rate to the environmental reservoir", as explained in more detail in Appendix A. This is now better reflected in the text of the methods section:

*"The first was the average rate at which humans are exposed **and contribute** to the environmental reservoir of infection (i.e., the overall transmission rate), ..."*

as well as in Table 2:

"Overall transmission rate (i.e., the annual rate ζ of exposure and contribution to the environmental reservoir)"

14e. The total population size is only described in the supplementary (~450 people at equilibrium), this should be included in the main text and justified. I would have expected a larger population to be modelled, or include some migration into the population.

Response 18

We now describe in the methods section (additions in bold): *"The model simulates the life histories of **about 450** individual humans in a **local** community (**the typical geographical scale of transmission**) and the life histories of the individual worms living within these humans, using time steps of one week."*

We acknowledge that adding the concept of multiple connected communities to our model (i.e., a meta-population model) would add another level of realism to our model, but would also further increase the complexity. Still, without implementing this, we can already say that due to migration, a meta-population model would predict lower probabilities of elimination and thus increase the opportunity for drug resistance to arise before elimination is achieved. We now mention this in the discussion section:

"We do note that our model only considers a single local community; if instead we would consider multiple communities between which people move, this would make transmission at low levels more stable [11,12] such that the probability of achieving elimination in all connected communities would be lower than we predict here, giving more opportunity for drug resistance to arise."

14f. Clarify that the worm lifespan follows a Weibull distribution.

Response 19

In Table 2, we now clarify (revisions in bold):

*"Ascaris lumbricoides (1-year average lifespan, mostly infecting children) and Necator americanus (3-year average lifespan, infecting both children and adults). **For both species, individual worm lifespans are assumed to follow a Weibull distribution (see Appendix A for details).**"*

14g. "Variation in drug efficacy between worms and treatments (95%-confidence interval (CI) of drug efficacy and corresponding σ on the log-odds scale)" - I don't understand this text and these values. Can they be presented on an absolute scale?

Response 20

We have simplified this to:

Left column: "*Variation in drug efficacy between worms and treatments*"

Right column: "*Standard deviation in log-odds of a worm being killed of $\sigma_{\text{trait}}^2 = 0.178$ ("low" variation, corresponding to a 95%-CI for probability of non-resistant worms being killed of 90.0–97.8%) or $\sigma_{\text{trait}}^2 = 1.044$ ("high" variation; 95%-CI of 80.0–99.5%).*"

14h. "90% reduction in drug efficacy in case of full resistance, and 40% reduction in drug efficacy in case of partial resistance [43]. This corresponds to 9.5% and 57% probability of a parasite being killed, respectively, compared to a 95% probability in case of absence of resistance." - this description is cumbersome. Consider simplifying to "Drug efficacy. Probability of a worm being killed is 95% in the absence of resistance, which drops to 57% or 9.5% in the case of partial and full resistance, respectively".

Response 21

Thank you for this suggestion. We have changed the description in Table 2 to: "*Drug efficacy: probability of a worm being killed is 95% in the absence of resistance, which drops to 57% (40% reduction) or 9.5% (90% reduction) in the case of partial and full resistance, respectively.*"

15. In general, more technical details from the supplementary should be included in the main text. It should be possible to understand the key features of the model from reading the main text alone.

Response 22

Thank you for raising this. To address this and a previous comment by the same reviewer about the main text not providing sufficient technical detail (point 14a above Response 14), we have added the following details from Table 2 and Appendix A to the methods section (additions in bold):

*"Here, we purposely focus on ~~hookworm~~ **N. americanus** and *A. lumbricoides* as benzimidazoles are not very efficacious against *T. trichiura* (~**50% egg reduction rate** [6])."*

*"Here we provide a high-level overview of the model; all technical details **and differences with previous models** are ~~provided~~ **described** in Appendix A."*

*"The model simulates the life histories of **about 450** individual humans in a **local** community (**the typical geographical scale of transmission**) and the life histories of the individual worms living within these humans, using time steps of one week (**which is appropriate given that the average lifespan of adult worms is 3 years for *N. americanus* and 1 year for *A. lumbricoides***)."*

“Humans contribute infective material to the reservoir as long as they are infected with female worms that produce fertilized eggs, which is only possible after a period of pre-patency (maturation **period** in the human host of **7–10 weeks**) ...”

“Further, the model explicitly simulates individual host participation to PC, accounting for age patterns in uptake and a degree of systematic non-participation. **Based on all of the above, the model predicts how drug efficacy translates into the effectiveness of PC to reduce STH infection levels in a human population.**”

“Thus, the actual probability of each worm being killed by drug treatment depends on their trait and the user-defined probability for a worm with trait $x_{\text{trait}} = 0$ to be killed. **In our simulations, we considered two levels of trait variation σ_{trait}^2 , such that in a worm population with a mean trait value of 0, the lower bound of a 95%-confidence interval for the probability of a worm being killed was 5 (“low” variation) or 10 percentage points (“high variation”) under the average drug efficacy (see Table 2 for details).**”

“The first was the average rate at which humans are exposed **and contribute** to the environmental reservoir of infection (i.e., the overall transmission rate), ...”

“...such that the majority of simulations represented settings with a baseline prevalence of infection of 10% to 60% **in the general population**, as measured by a single Kato-Katz faecal smear...”

“The third transmission-related parameter was the suitability of the environment for survival of eggs or larvae (i.e., the average lifespan of eggs or larvae), which was allowed to vary between half and twice the values reported in literature (**average of 2–4 weeks for N. americanus, 1.5 months for A. lumbricoides** [7,8]) to capture potential geographical variation.”

“~~All~~ **Per simulation, each of the three parameter values for transmission conditions were** ~~was~~ drawn from a uniform distributions on the logarithmic scale.”

“That is, the initial distribution of quantitative traits was assumed to follow a normal distribution with mean zero and variance σ_{trait}^2 (**either low or high variation, as described above**).”

16. The discussion is lengthy relative to the results section and could be condensed.

Response 23

To reduce the size of the discussion, we propose the following candidate sections to be removed, as they are more peripheral to the main focus of the paper (i.e., the origins of resistance-conferring SNPs, and discussion of human filarial infections). With these sections removed, the discussion is now ~1520 words long (including new material to address other reviewer comments).

REMOVED [189 words]: “In our study, we assumed that resistance-conferring alleles are already present in the population at the start of PC and that said populations had not necessarily been exposed to PC before. As such, we did not make any explicit assumptions about how those alleles came to be present. According to Skuce et al [13], there are three processes by which resistance-conferring SNPs could arise in a helminth parasite population. Such SNPs might (1) originate from a single mutation (i.e., in a single worm) that appeared and eventually swept through the parasite population while under selection pressure, (2) be always present as part of standing genetic variation in

parasite populations, or (3) repeatedly appear by recurrent spontaneous mutations. Skuce et al argue that for benzimidazole resistance in *Teladorsagia circumcincta* in sheep, the third hypothesis is the most likely based on the observed amount and pattern of genetic diversity. Although we do not explicitly simulate spontaneous mutations, our results are still valid as once resistance-conferring SNPs are sufficiently established (as in our simulations), the population dynamics of resistance are mostly dictated by PC and further spontaneous mutations do not matter.”

REMOVED [238 words]: “As for lymphatic filariasis and onchocerciasis in humans, two other human helminth infections controlled by PC, the lifespan of adult parasites is also much higher (6–10 years) than for STH [14], slowing down emergence of resistance. Further, the drugs used in PC (also) kill microfilarial life stages in hosts, greatly reducing transmission between PC rounds. In the case of lymphatic filariasis, PC usually involves treatment with two drugs from different drug families, which further decreases the risk and speed of drug resistance emerging and makes it less likely that resistance will impact the effectiveness of control programs [15]. Regarding onchocerciasis, there have been several reports of reduced genetic variation and shifts in allele frequency following ivermectin treatment [16–20]. It has been suggested that drug resistance in onchocerciasis is polygenic [21], meaning that its spread would be relatively slow. However, the authors based their analysis on a draft reference genome of *Onchocerca volvulus*, rather than a genome that was fully assembled at the chromosomal scale. Studies in *Haemonchus contortus* have shown that using a draft reference genome can lead to an artificial impression of polygenic inheritance due to the scattering of the selection signal across multiple scaffolds [22,23]. As such, the inheritance mechanism in onchocerciasis remains unclear. Last, more generally, as long as resistance in filariases only manifests as an adult worm trait, the microfilaricidal effects of PC will slow down the spread of any resistance [15].”

Where possible, we have also tried to condense the existing text in the discussion section, although this was with limited effect.

17. Data availability - the authors should make the "model generated data" available from a repository and it should be possible to reproduce the figures from the code and data. There is no reason to restrict access to synthetic data.

Response 24

We have made all model-generated data available through the *figshare* server via the manuscript submission process.

Reviewer #2

The manuscript is well-structured and well-penned, offering significant insights into the parameterization of hookworm (*Necator americanus*) and *Ascaris*, drawn meticulously from extensive literature on their biology and epidemiology. The authors have proficiently incorporated means to represent varied resistance genotypes encompassing both singular and multiple genes within the simulation. The simulation harbors a profound public health message, emphasizing the substantial risk and rapid evolution of benzimidazole resistance in Soil-Transmitted Helminths (STH), thereby underscoring the immediate need for the development of vigilant surveillance and responsive mitigation strategies. However, there are critical elements in the model that necessitate addressal, hindering its immediate acceptance for publication.

1. Lack of Model Validations:

The manuscript lacks validations for the newly introduced model. Even though this model builds upon previous works on worm infection simulation, it demands re-validation against clinical aspects of the diseases. The validations may include the following points but not limit to:

- Worm distribution's variability with age and transmission intensity and supporting data for the same. For instance, Figure B1 suggests a worm population size of over 10,000 worms, approximating to 25 worms per individual. Validation data for this is imperative.
- Corroboration of the age-stratified prevalence at equilibrium with existing data or specific demographics. How the shape changing on different transmission intensity.
- Examination of the relationship between prevalence and the transmission rate.
- Validation of the prevalence of egg presence in feces.

Response 25

The reviewer is asking for a validation of the entire model, whereas the backbone of our STH model (i.e., structure and parameters for transmission and impact of PC) is based on almost 10 years of published work, including model validations. This was perhaps not clear enough from the manuscript. We now try to better clarify in the methods section:

*“WORMSIM predictions for STH have been previously validated against field data on trends in *N. americanus* and *A. lumbricoides* infection levels before and during PC [24,25]. As such, many **In the new model, basic assumptions about parasite biology and impact of PC in the new model are identical to WORMSIM, such that the new model (with drug resistance turned off) reproduced the behaviour of WORMSIM.**”*

The validations requested by the reviewer have been previously done, as far as possible. The most important of these validations are with regard to age patterns in infection levels, the distribution of infection intensities across individuals, and the impact of PC on trends in infection levels in the population [24,25]. However, some of the aspects that the reviewer raises cannot be fully validated because this would require data that cannot (easily) be collected:

Worm population size: unlike in vector-borne infectious diseases such as malaria where blood samples can be easily tested, in STH, parasite loads are typically measured by proxy in terms of faecal egg densities. Worm expulsion studies to count actual adult worms are rarely performed as they are extremely difficult and labour intensive; try finding a hair-thin <1cm-long hookworm in several containers of faeces (per tested person) without breaking the worms. As such, all existing STH transmission models rely on the association between faecal egg counts and adult worm loads as found in a series of historical expulsion studies (e.g., [26]): ~200 eggs per gram faeces per female worm for *N. americanus* and ~9 thousand eggs per gram faeces per female worm for *A. lumbricoides*. These data also inform the degree of density-dependence in worm fecundity, which leads to the typical tapering off of faecal egg densities as the number of female worms increases. Given these associations and the levels in faecal egg counts observed in the field (zero to in the order of many thousands), an average load of 25 worms per person is not uncommon.

Relationship between infection levels and the transmission rate: unlike for vector-borne diseases, for STH it is not possible to quantify this relationship. For instance, in malaria, it is possible to quantify the presence of malaria parasites in mosquitoes and combine this with estimates of mosquito density and the mosquito-human contact rate to directly inform malaria transmission models. For STH, the parasite load in the environmental reservoir

cannot be reliably measured (not for lack of trying) and human behaviour in terms of exposure and contribution to this reservoir is even harder to measure. As such, the absolute magnitude of the environmental reservoir is completely unidentified for STH; we only know – again from historical experiments – the rate at which eggs and larvae die in the environment. As such, in STH models, the average transmission rate and degree of exposure heterogeneity are calibrated to reproduce observed population-level distributions of egg counts. These aspects, including age patterns in transmission, have been investigated and validated in previous publications by our own group [24,25] and others [27–30].

Of course, we acknowledge that there may well be more variation / uncertainty in transmission condition than we can derive from the aforementioned historical studies. Therefore, we performed our simulation for a wide range of transmission conditions, and show how this variation/uncertainty affects prospects of elimination but only marginally influences trends in declining drug efficacy.

2. Population Size:

A human population size of 400 appears insufficient to accurately simulate drug resistance in an individual-based simulation. Especially considering a 5-10% prevalence and 40% of the population aged 0-15, the resultant 16 individuals harboring parasites provide a limited view of the system dynamics when PC targets only these individuals.

Response 26

We understand the reviewer's concern, but note that fortunately, the simulated prevalences of infection were much higher (20%-80% of individuals had at least one female worm, corresponding to 10%-60% egg positivity) and these prevalences pertained to the general population (all ages). We now better clarify this in the methods text and Table 2:

*"...corresponding to a 10% to 60% prevalence of infection **in the general population**, as measured by a single Kato-Katz faecal smear".*

As for the population size of 400: small rural communities are the typical geographical scale at which STH transmission takes place, which we now better clarify:

*"The model simulates the life histories of **about 450 individual humans in a local community (the typical geographical scale of transmission)**"*

In response to an earlier comment by another reviewer (Response 18), we now also state in the discussion:

"We do note that our model only considers a single local community; if instead we would consider multiple communities between which people move, this would make transmission at low levels more stable [11,12] such that the probability of achieving elimination in all connected communities would be lower than we predict here, giving more opportunity for drug resistance to arise."

3. Role of Immunity:

Immunity is pivotal in transmission dynamics and, presumably being age-dependent, could potentially influence the infection duration and the contribution to the Force of Infection (FOI) in adults despite equivalent environmental exposure levels.

Response 27

Given that most helminths are masters in evading the host immune system, in many helminth infections, acquired immunity plays a rather unusual role. For soil-transmitted helminths, faecal expulsion data (i.e., the association between number of female worms and faecal egg density) suggests a role for the host immune system in the regulation of worm fecundity (egg output per worm). The resulting negative density dependence in worm fecundity is explicitly captured in our model – see equation 9 in Appendix A. If we were to consider a potential additional role of immunity in STH transmission (e.g., via parasite establishment) and if there were data to inform this, this would not change the message of our paper as waning immunity would make elimination even harder and allow more time for resistance to emerge.

As for age patterns in STH infection levels, these are largely driven by the transmission route (oral ingestion of contaminated soil for *A. lumbricoides*; barefoot walking and skin exposure to free-living larvae for *N. americanus*) and the associated age patterns in exposure.

The above notions/assumptions are widely accepted since the founding work by Anderson and May in the 1980s [31,32] and are implemented in all existing mathematical models for transmission of STH and schistosomiasis.

4. Cost of Resistance:

There is a prevailing hypothesis of an inherent cost to resistance, potentially impacting the number of eggs per generation or the lifespan of the resistance genotype. Any model addressing drug resistance should incorporate, at minimum, a foundational form and sensitivity analysis reflecting the potential impacts of these resistance costs.

Response 28

We agree with the reviewer that in general, a fitness loss makes sense biologically and evolutionarily. We have performed a sensitivity analysis for the impact of a fitness loss in a scenario with monogenic resistance (recessive, co-dominant, and dominant) during annual community-based PC. The fitness loss was defined as a reduction in the egg output of female worms, expressed as the relative egg productivity for particular genotypes (1 = no fitness loss):

	aa	aA	AA
Recessive	1	1	2/3
Co-dominant	1	5/6	2/3
Dominant	1	2/3	2/3

This resulted in a modest but noticeable impact on both elimination (detailed figure in Appendix E) and the speed of decline in drug efficacy, although not to a degree that should make us less concerned about resistance:

These new simulation results are now included in Appendix E (sensitivity analyses) and are briefly referred to in the results section (second-to-last paragraph) and the discussion (first paragraph):

Results (entire new sentence): *“Fifth and last, a fitness loss in terms of lower female worm egg productivity (a 1/3 reduction for full resistance and a 1/6 reduction for partial resistance) resulted in slightly higher probability of elimination and a slower decline in drug efficacy when elimination was not achieved (Appendix E, Figures E9-10).”*

Discussion (new material in bold): *“Given the marked differences in risk and speed of emergence of drug resistance across different genetic mechanisms, there is a need to better understand the main inheritance mechanisms for potential drug resistance in STH **and a potential associated fitness loss (although the latter would probably have a modest impact).**”*

As there is no clear evidence of a fitness cost of drug resistance in veterinary or human helminth infections [33,34], we have not further changed the main message of our paper.

5. Drug Impact on Eggs:

The drugs utilized in Mass Drug Administration (MDA) serve not only to exterminate the parasites but also to alter the egg-positive rate, both directly and indirectly as delineated in the literature. Thus, validation regarding the egg-positive rates pre and post-MDA is vital for an accurate drug model.

Response 29

The impact of PC on faecal egg counts operates exclusively via the drug treatment’s effect on the adult worms and the ensuing change in worm loads and density-dependent worm fecundity. These processes are explicitly captured in the model. As explained in Response 25, the model-predicted impact of PC on infection levels (in terms of egg counts and prevalence) have been previously validated [24,25], which is now hopefully also clearer from the methods section.

Minor Concerns:

- The warm-up period of 300 years appears excessive; a justification for this duration is needed.
- An explanation for opting for a weekly timestep over a daily one for the simulation is required.
- Clarification is needed on the reproductive capabilities of a male worm during multiple infections, specifically whether a long-infected male worm can still reproduce by mating with a newly infected female one.

Response 30

Warm-up duration: as explained in Appendix A, the 300-year warm-up consists of two components: 250 years for human demography (which we warm up in monthly time steps) and 50 years for worm demography (which we warm up in weekly time steps). The 250-year component is appropriate given the maximum human lifespan of 90 years; the 50-year component is of appropriate length given the average parasite lifespan of 1–3 years and the speed of the parasite population dynamics.

Simulation time step: we now clarify in the methods section *“The model simulates the life histories of **about 450** individual humans in a **local** community (**the typical geographical scale of transmission**) and the life histories of the individual worms living within these humans, using time steps of one week (**which is appropriate given that the lifespan of adult worms is 3 years for *N. americanus* and 1 year for *A. lumbricoides***).”*

Reproduction: as explained in Appendix A and implemented in all existing mathematical models for STH transmission, STH are considered highly promiscuous parasites and it is assumed that for all female worms to be inseminated, only a single male worm is required (of any age, although their lifespan is finite).

While the manuscript is commendable for its organization, clarity, and significant public health implications, addressing the highlighted concerns is essential to enhancing its overall validity and relevance. The resolution of issues related to model validation, consideration of immunity, population size, and cost of resistance will considerably strengthen the credibility and contribution of this work to the scientific community.

Reviewer #3

This work is of genuine importance, for as the authors point out, little has been done to date examining the risk of emerging resistance on the control of STH infections globally, or within endemic communities. The absence of standardized or easily applied methods to monitor for genetically mediated resistance in human helminth infections raises concerns about the ability to detect relevant reductions in treatment response, especially in the setting of targeted chemotherapy or mass drug administration programs. That said, there are some aspects of the manuscript that would benefit from greater clarity and potentially analysis.

1. The authors estimate a baseline “efficacy” of 95% for albendazole against both *Ascaris* and *Necator* (Fig 1, Fig 4). It’s not clear if that refers to cure rate (CR) or Egg Reduction Rate (ERR), which are two useful but clearly distinct metrics for measuring drug response.

Response 31

Thank you for raising that this was unclear. In the model itself, the process behind drug efficacy (as one would observe it in terms of ERR or CR) is defined in terms of the proportion of worms killed. The choice of 95% efficacy was based on ERRs from field studies [6,35], which we now better clarify in the methods section:

*“95% of worms killed, **which based on observed egg reduction rates, is a plausible value [6,35] and was chosen for illustrative purposes”***

In response to one of the other reviewer comments (Response 1), we now also illustrate how ERRs based on Kato-Katz egg counts might be expected to evolve over time in case of resistance. We do not consider CRs as these should not be used in drug efficacy

surveillance due to being heavily confounded by pre-treatment infection levels (see also Response 1).

2. It is well recognized that the benzimidazole drugs do not have the same activity against the three major STHs (Ascaris, Trichuris, hookworms), and that there are intrinsic differences that should be taken into account. Given that cure rates for Ascaris are typically much higher than those reported for hookworm, accounting for the difference might impact the models presented. The authors should comment on whether these have been considered.

Response 32

We agree with the reviewer: we also initially thought that these differences in drug efficacy against different STH species would be important. We therefore performed sensitivity analyses for alternative assumptions about drug efficacy: 80%, 90%, 95% (main analysis), and 99%. This range was chosen as being relevant to *A. lumbricoides* and hookworm species, based on literature [6,35].

It turned out that this aspect did not have much of an impact on trends in declining drug. This was previously illustrated in Figure 4, which has now been moved to Appendix 4 based on other reviewer comments. We therefore have added a textual emphasis in the results section about the lowest efficacy that we considered (new text in bold): *“Finally, lower drug efficacy (80%) against non-resistant worms (using monogenic resistance as an example) led to somewhat lower probability of elimination but also somewhat slower declines in drug efficacy if elimination was not achieved (Appendix E, Figures E5-6).”*

With regard to trichuriasis, in the first paragraph of the methods section now more clearly states that we do not cover this here as benzimidazole efficacy is already very low against *Trichuris trichiura* (new material in bold): *“as benzimidazoles are not very efficacious against *T. trichiura* (~50% egg reduction rate [6])”*.

3. It is worth distinguishing between “efficacy” and “effectiveness,” the latter being how well the drug works in regular practice, as opposed to how it works in the setting of a controlled trial.

Response 33

Thank you for raising this important point. In our model, PC is defined in terms of (1) treatment efficacy at the individual level (proportion of non-resistant worms killed), (2) the impact of resistance-conferring alleles on the proportion of worms killed, and (3) patterns in individual uptake of PC (age patterns, individual systematic (non-)participation). Based on these three aspects, combined with the STH population dynamics (e.g., pre-patent worms becoming patent shortly after treatment), the model simulates the effectiveness of PC in terms of reducing infection levels in individuals and the population. In response to the reviewer’s comment, we have carefully checked our manuscript on correct use of “efficacy” and “effectiveness”. In the methods section, we have now added a sentence:

“Based on all of the above, the model predicts how drug efficacy translates into the effectiveness of PC to reduce STH infection levels in a human population.”

4. The authors reference studies showing host factors that may alter response to treatment, yet it’s not clear from the analysis that this was accounted for.

Response 34

Thank you for raising this. Indeed, in the introduction we mention that estimates of drug efficacy are confounded by host-level factors such as variation in benzimidazole pharmacokinetics [2], variation in pre-treatment infection levels [35–37], density-dependent worm fecundity [38], and nutritional factors [39].

In our model, we explicitly capture the impact of pre-treatment infection levels and density-dependent worm fecundity on the effect of treatment. Variation in pharmacokinetics and nutritional factors are not an explicit part of the model as host-level variation due to these factors was considered to be minor or negligible compared to all of the other sources of individual heterogeneity in the model (exposure levels, worm loads, PC uptake, stochastic treatment effects).

In the polygenic resistance model, we do consider random temporal variation in drug efficacy (as a counterpart to the variation in drug efficacy due to worm genetics), which could be considered to include nutritional and (random) pharmacokinetic noise (at the worm-level). For monogenic resistance, however, we assume that drug efficacy is fully dictated by the worm genotype. Still, in both processes (polygenic and monogenic resistance), further noise is added to the treatment effects by the stochastic process of individual worms surviving or being killed by treatment. As such, at the population level, we reason that additional variation due to pharmacokinetics and nutritional factors is indirectly captured in the random noise of treatment effects and PC uptake.

In Appendix A, section “Parasitological effects of drug treatment”, we have now added:

“Note that we do not explicitly consider (host-level) variation in treatment effects due to pharmacokinetics or nutritional factors (apart from random (worm-level) noise in treatment effects in the polygenic resistance mechanics). Although such effects can be important in clinical and experimental studies, here we assume them to be minor or negligible compared to all of the other sources of individual heterogeneity in the model (exposure levels, worm loads, PC uptake, stochastic treatment effects).”

5. The authors use a baseline prevalence of approximately 60% as a starting point for the model. However, this is a very high prevalence and probably not the case in most endemic communities today, since over the past 2 decades endemic communities are frequently characterized by low prevalence (20-30%) and low intensity infections. It would be helpful if the authors commented on how the baseline prevalence chosen might influence the outcome of the predictive modeling they have carried out.

Response 35

The baseline prevalence of approximately 60% (i.e., people carrying at least one female worm) is indeed the average of the simulations that we highlight in Figure 1 (red solid line). This average is based on a set of 1,000 repeated simulations (the transparent black and grey lines in Figure 1) across a range of transmission conditions such that the baseline prevalence varied significantly across simulations: between approximately 20% and 80% (10%-60% in terms of egg-positive individuals, see Appendix B). And indeed, as our model predicts, infection levels do come down with PC over the years. Later on, in the fourth paragraph of the results section and with reference to Appendix D, we discuss how stratifying the analysis by transmission conditions (including baseline infection levels) significantly affects prospects of elimination but hardly affect trends in declining drug resistance in case elimination is not achieved.

References

1. Coffeng LE, Vlamincck J, Cools P, Denwood M, Albonico M, et al. (2023) A general framework to support cost-efficient fecal egg count methods and study design choices for large-scale STH deworming programs-monitoring of therapeutic drug efficacy as a case study. *PLoS Negl Trop Dis* **17**: e0011071.
2. Vercruyssen J, Albonico M, Behnke JM, Kotze AC, Prichard RK, et al. (2011) Is anthelmintic resistance a concern for the control of human soil-transmitted helminths? *Int J Parasitol Drugs Drug Resist* **1**: 14–27.
3. World Health Organization (2013) Assessing the efficacy of anthelmintic drugs against schistosomiasis and soil-transmitted helminthiasis. Geneva. 29 p.
4. Coffeng LE, Levecke B, Hattendorf J, Walker M, Denwood MJ (2021) Survey design to monitor drug efficacy for the control of soil-transmitted helminthiasis and schistosomiasis. *Clin Infect Dis* **72**: S195–S202.
5. Coffeng LE, Stolk WA, Bakker R, de Vlas SJ (2016) Soil-transmitted helminth (STH) control, elimination, and development of drug resistance: repercussions of systematic non-participation to preventive chemotherapy. *65th Annual Meeting of the American Society of Tropical Medicine & Hygiene*. Atlanta: American Society of Tropical Medicine & Hygiene. pp. 221–222.
6. Vercruyssen J, Behnke JM, Albonico M, Ame SM, Angebault C, et al. (2011) Assessment of the anthelmintic efficacy of albendazole in school children in seven countries where soil-transmitted helminths are endemic. *PLoS Negl Trop Dis* **5**: e948.
7. Jourdan PM, Lamberton PHL, Fenwick A, Addiss DG (2018) Soil-transmitted helminth infections. *Lancet* **391**: 252–265.
8. Bethony J, Brooker S, Albonico M, Geiger SM, Loukas A, et al. (2006) Soil-transmitted helminth infections: ascariasis, trichuriasis, and hookworm. *Lancet* **367**: 1521–1532.
9. World Health Organization (2017) Preventive chemotherapy to control soil-transmitted helminth infections in at-risk population groups. Geneva.
10. Anderson RM, Truscott J, Hollingsworth TD (2014) The coverage and frequency of mass drug administration required to eliminate persistent transmission of soil-transmitted helminths. *Philos Trans R Soc Lond B Biol Sci* **369**: 20130435.
11. de Vos AS, Stolk WA, de Vlas SJ, Coffeng LE (2018) The effect of assortative mixing on stability of low helminth transmission levels and on the impact of mass drug administration: Model explorations for onchocerciasis. *PLoS Negl Trop Dis* **12**: e0006624.
12. de Vos AS, Stolk WA, Coffeng LE, de Vlas SJ (2021) The impact of mass drug administration expansion to low onchocerciasis prevalence settings in case of connected villages. *PLoS Negl Trop Dis* **15**: e0009011.
13. Skuce P, Stenhouse L, Jackson F, Hypsa V, Gilleard J (2010) Benzimidazole resistance allele haplotype diversity in United Kingdom isolates of *Teladorsagia circumcincta* supports a hypothesis of multiple origins of resistance by recurrent mutation. *Int J Parasitol* **40**: 1247–1255.

14. Taylor MJ, Hoerauf A, Bockarie M (2010) Lymphatic filariasis and onchocerciasis. *Lancet* **376**: 1175–1185.
15. Schwab AE, Churcher TS, Schwab AJ, Basáñez M-G, Prichard RK (2006) Population genetics of concurrent selection with albendazole and ivermectin or diethylcarbamazine on the possible spread of albendazole resistance in *Wuchereria bancrofti*. *Parasitology* **133**: 589–601.
16. Ardelli BF, Prichard RK (2007) Reduced genetic variation of an *Onchocerca volvulus* ABC transporter gene following treatment with ivermectin. *Trans R Soc Trop Med Hyg* **101**: 1223–1232.
17. Bourguinat C, Pion SDS, Kamgno J, Gardon J, Gardon-Wendel N, et al. (2006) Genetic polymorphism of the beta-tubulin gene of *Onchocerca volvulus* in ivermectin naïve patients from Cameroon, and its relationship with fertility of the worms. *Parasitology* **132**: 255–262.
18. Bourguinat C, Pion SDS, Kamgno J, Gardon J, Duke BOL, et al. (2007) Genetic selection of low fertile *Onchocerca volvulus* by ivermectin treatment. *PLoS Negl Trop Dis* **1**: e72.
19. Bourguinat C, Ardelli BF, Pion SDS, Kamgno J, Gardon J, et al. (2008) P-glycoprotein-like protein, a possible genetic marker for ivermectin resistance selection in *Onchocerca volvulus*. *Mol Biochem Parasitol* **158**: 101–111.
20. Nana-Djeunga H, Bourguinat C, Pion SDS, Kamgno J, Gardon J, et al. (2012) Single nucleotide polymorphisms in β -tubulin selected in *Onchocerca volvulus* following repeated ivermectin treatment: possible indication of resistance selection. *Mol Biochem Parasitol* **185**: 10–18.
21. Doyle SR, Bourguinat C, Nana-Djeunga HC, Kengne-Ouafo JA, Pion SDS, et al. (2017) Genome-wide analysis of ivermectin response by *Onchocerca volvulus* reveals that genetic drift and soft selective sweeps contribute to loss of drug sensitivity. *PLoS Negl Trop Dis* **11**: e0005816.
22. Doyle SR, Illingworth CJR, Laing R, Bartley DJ, Redman E, et al. (2019) Population genomic and evolutionary modelling analyses reveal a single major QTL for ivermectin drug resistance in the pathogenic nematode, *Haemonchus contortus*. *BMC Genomics* **20**: 218.
23. Wit J, Workentine ML, Redman E, Laing R, Stevens L, et al. (2022) Genomic signatures of selection associated with benzimidazole drug treatments in *Haemonchus contortus* field populations. *Int J Parasitol* **52**: 677–689.
24. Coffeng LE, Bakker R, Montresor A, de Vlas SJ (2015) Feasibility of controlling hookworm infection through preventive chemotherapy: a simulation study using the individual-based WORMSIM modelling framework. *Parasit Vectors* **8**: 541.
25. Coffeng LE, Truscott JE, Farrell SH, Turner HC, Sarkar R, et al. (2017) Comparison and validation of two mathematical models for the impact of mass drug administration on *Ascaris lumbricoides* and hookworm infection. *Epidemics* **18**: 38–47.
26. Anderson RM, Schad GA (1985) Hookworm burdens and faecal egg counts: an analysis of the biological basis of variation. *Trans R Soc Trop Med Hyg* **79**: 812–825.

27. Truscott JE, Turner HC, Farrell SH, Anderson RM (2016) Soil-Transmitted Helminths: Mathematical Models of Transmission, the Impact of Mass Drug Administration and Transmission Elimination Criteria. *Adv Parasitol* **94**: 133–198.
28. Anderson RM, Truscott JE, Pullan RL, Brooker SJ, Hollingsworth TD (2013) How effective is school-based deworming for the community-wide control of soil-transmitted helminths? *PLoS Negl Trop Dis* **7**: e2027.
29. Truscott J, Hollingsworth TD, Anderson RM (2014) Modeling the interruption of the transmission of soil-transmitted helminths by repeated mass chemotherapy of school-age children. *PLoS Negl Trop Dis* **8**: e3323.
30. Farrell SH, Coffeng LE, Truscott JE, Werkman M, Toor J, et al. (2018) Investigating the Effectiveness of Current and Modified World Health Organization Guidelines for the Control of Soil-Transmitted Helminth Infections. *Clin Infect Dis* **66**: S253–S259.
31. Anderson RM, May RM (1985) Helminth infections of humans: mathematical models, population dynamics, and control. *Adv Parasitol* **24**: 1–101.
32. Anderson RM, May RM (1991) *Infectious Diseases of Humans: Dynamics and Control*. Oxford & New York: Oxford University Press.
33. Conder GA, Campbell WC (1995) Chemotherapy of nematode infections of veterinary importance, with special reference to drug resistance. *Adv Parasitol* **35**: 1–84.
34. Nielsen MK, Reinemeyer CR, Donecker JM, Leathwick DM, Marchiondo AA, et al. (2014) Anthelmintic resistance in equine parasites--current evidence and knowledge gaps. *Vet Parasitol* **204**: 55–63.
35. Levecke B, Montresor A, Albonico M, Ame SM, Behnke JM, et al. (2014) Assessment of anthelmintic efficacy of mebendazole in school children in six countries where soil-transmitted helminths are endemic. *PLoS Negl Trop Dis* **8**: e3204.
36. Levecke B, Mekonnen Z, Albonico M, Vercruysse J (2012) The impact of baseline faecal egg counts on the efficacy of single-dose albendazole against *Trichuris trichiura*. *Trans R Soc Trop Med Hyg* **106**: 128–130.
37. Bennett A, Guyatt H (2000) Reducing intestinal nematode infection: efficacy of albendazole and mebendazole. *Parasitol Today* **16**: 71–74.
38. Kotze AC, Kopp SR (2008) The potential impact of density dependent fecundity on the use of the faecal egg count reduction test for detecting drug resistance in human hookworms. *PLoS Negl Trop Dis* **2**: e297.
39. Humphries D, Nguyen S, Kumar S, Quagraine JE, Otchere J, et al. (2017) Effectiveness of Albendazole for Hookworm Varies Widely by Community and Correlates with Nutritional Factors: A Cross-Sectional Study of School-Age Children in Ghana. *Am J Trop Med Hyg* **96**: 347–354.

REVIEWERS' COMMENTS

Reviewer #1 (Remarks to the Author):

The authors have responded thoughtfully to the suggestions posed by reviewers. I am particularly pleased to see the additional analysis on ERR and further sensitivity analyses.

My only remaining comment relates to the new Figure 5. I would advise adding an uncertainty interval (e.g. with shading) around "Probability (%) of ERR <85%" (blue / cyan line) to better understand the variability in simulated outputs. I also suggest removing the purple line from this plot ("Probability of true drug efficacy <90% if ERR <85%") as this doesn't improve understanding and shows noisy output.

After resolution of this final comment, I am happy to recommend the study for publication in Nature Comms.

Reviewer #2 (Remarks to the Author):

Dear Authors,

Firstly, I would like to express my agreement and appreciation for your detailed and insightful response to the points raised in the previous review. It is evident that considerable effort has been put into addressing and clarifying most of the concerns.

However, there remain specific aspects that warrant further clarification. Notably, the difference in programming paradigms between WORMSIM and simresist. While WORMSIM is developed in Java utilizing an object-oriented programming approach, simresist is constructed in R with a focus on procedural and table-based methods. This fundamental difference in architecture, despite simresist's reliance on assumptions, formulas, and mechanisms derived from WORMSIM, necessitates a re-validation of simresist in comparison to WORMSIM. Such validation is crucial to establish the integrity of the newly implemented model in simresist.

A practical approach for this validation could involve the extraction of key output features from WORMSIM and attempting to replicate them in simresist. Essentially, under identical conditions (specifically with the drug resistance feature disabled), both simresist and WORMSIM should yield comparable results. This parallel analysis would provide a robust basis for confirming the validity and reliability of simresist as an independent model.

I look forward to your thoughts and further developments in this area.

Best regards,

Tran Dang Nguyen
Research Assistant Professor
The Pennsylvania State University

Reviewer #3 (Remarks to the Author):

The authors have adequately addressed the comments raised from review of the previous submission. There are no further changes suggested.

Response to second set of reviewer comments

We thank the reviewers for their positive and constructive comments.

Reviewer #1

The authors have responded thoughtfully to the suggestions posed by reviewers. I am particularly pleased to see the additional analysis on ERR and further sensitivity analyses.

My only remaining comment relates to the new Figure 5. I would advise adding an uncertainty interval (e.g. with shading) around "Probability (%) of ERR <85%" (blue / cyan line) to better understand the variability in simulated outputs. I also suggest removing the purple line from this plot ("Probability of true drug efficacy <90% if ERR <85%") as this doesn't improve understanding and shows noisy output.

After resolution of this final comment, I am happy to recommend the study for publication in Nature Comms.

Response 1

We have revised Figure 5 as suggested, removing the purple line (positive predictive value of ERR<85%, which is now only shown in Appendix F) and adding a measure of uncertainty for the probability of finding an ERR <85%. Given the highly skewed variability of ERR survey results, we have plotted the median and IQR of the simulated probabilities. The IQR is wider for ascaris than for hookworms as ascaris infections are associated with much higher overdispersion of repeated egg counts in the same individual.

Reviewer #2

Firstly, I would like to express my agreement and appreciation for your detailed and insightful response to the points raised in the previous review. It is evident that considerable effort has been put into addressing and clarifying most of the concerns.

However, there remain specific aspects that warrant further clarification. Notably, the difference in programming paradigms between WORMSIM and simresist. While WORMSIM is

developed in Java utilizing an object-oriented programming approach, simresist is constructed in R with a focus on procedural and table-based methods. This fundamental difference in architecture, despite simresist's reliance on assumptions, formulas, and mechanisms derived from WORMSIM, necessitates a re-validation of simresist in comparison to WORMSIM. Such validation is crucial to establish the integrity of the newly implemented model in simresist.

A practical approach for this validation could involve the extraction of key output features from WORMSIM and attempting to replicate them in simresist. Essentially, under identical conditions (specifically with the drug resistance feature disabled), both simresist and WORMSIM should yield comparable results. This parallel analysis would provide a robust basis for confirming the validity and reliability of simresist as an independent model.

Response 2

As requested, in Appendix A we now show a figure (Figure A12) comparing trends in infection as predicted by WORMSIM and the simresist model (in absence of drug resistance), demonstrating that simresist reproduces the original WORMSIM model behaviour.

Reviewer #3

The authors have adequately addressed the comments raised from review of the previous submission. There are no further changes suggested.